

# Life cycle of bamboo in southwestern Amazon and its relation to fire events

Ricardo Dalagnol[1], Fabien Hubert Wagner[1], Lênio Soares Galvão[1], Bruce Walker Nelson[2], and Luiz Eduardo Oliveira e Cruz de Aragão[1]

[1]Remote Sensing Division, National Institute for Space Research - INPE, São José dos Campos, SP, 12227-010, Brazil
[2]Environmental Dynamics Department, National Institute of Amazonian Research - INPA, Manaus, AM, 69067-375, Brazil

**Correspondence:** Ricardo Dalagnol (ricds@hotmail.com)

**Abstract.** Bamboo-dominated forests comprise 1% of the world's forests. In southwest Amazon, a prior investigation mapped 16.15 million ha of bamboo-dominated forests by visual interpretation of Landsat and MODIS satellite images. They observed that the near infrared (NIR) wavelength was important to discriminate adult and dead bamboo areas, and estimated an average life cycle of 28 years for the *Guadua spp.* bamboo that dominate the region. In these bamboo areas, flowering and fruiting

occur once in a lifetime before death, which produce massive quantities of necromass. The 'bamboo-fire hypothesis' argues that increased dry fuel after die-off enhances fire probability, creating opportunities that favor bamboo. In this study, we developed and validated a method to map the bamboo die-off and its spatial distribution using satellite-derived time series of vegetation reflectance from MODIS (MAIAC) data and explored the fire hypothesis by evaluating the relationship between bamboo die-off and fires detected by MODIS thermal anomalies product in southwest Amazon. Our findings show that the

NIR is the most sensitive spectral interval to characterize bamboo growth and cohort age. Automatic detection of historical bamboo die-off achieved an accuracy above 79%. We mapped and estimated 15.5 million ha of bamboo-dominated forests in the region. The mean patch size during 2001-2017 die-off events was 85 km$^2$, while the largest patch covered up to 6162 km$^2$. The 'bamboo-fire hypothesis' was not fully supported, because most bamboo cohorts did not burn after die-off. Nonetheless, fire occurrence was 45% higher in dead than live bamboo in drought years, associated with ignition sources from land use,

suggesting a bamboo-human-fire association. Since fire favors bamboo development, this may contribute to the maintenance of high bamboo density where it is already present, the expansion of bamboo into adjacent bamboo-free forests, or even bring deadly consequences to these adjacent forests under climate change effects.

## 1 Introduction

Bamboo-dominated forests represent 1% of global forests. They occur in tropical, subtropical and mild temperate zones and

are found mainly in Asia (24 million ha), South America (10 million ha) and Africa (2.8 million ha) (Lobovikov et al., 2007). Their spatial distribution is likely underestimated in South America as a recent study showed that these forests cover at least 16.15 million ha of Amazonian forests over Brazil, Peru and Bolivia (Carvalho et al., 2013).



Bamboo is a major forest product that plays an important economic and cultural role in the Amazon. It has been used for over a millennium by indigenous people for shelter, food, fuel, hunting, fishing, and musical instruments (Lobovikov et al., 2007; Rockwell et al., 2014). The first studies on the distribution of these forests in the Amazon region postulated that they occurred as a consequence of human disturbance or were deliberately planted (Sombroek, 1966; Balée, 1989). However, recent

phytolith analysis revealed that bamboo-dominated these forests before human occupation in South America (Olivier et al., 2009; Watling et al., 2017).

In the southwest Amazon, the predominant forest type is non-flooded open-canopy rain forest on terra firme, often dominated by *Guadua* bamboos and mostly (93%) preserved (IBGE, 2006; Trancoso et al., 2010). In bamboo-dominated areas, two species of semi-scandent woody bamboos predominate: *Guadua weberbaueri* Pilger and *Guadua sarcocarpa* Londoño &

Peterson. Like many other woody bamboo species, these Guadua bamboos are semelparous, producing flowers and fruits once in a lifetime before dying (Janzen, 1976; Griscom and P. Mark S. Ashton, 2003). Flowering, fruiting and death can be massive and highly synchronized in space and time. Their diameter at breast height (DBH) ranges from 4 to 24 cm (Castro et al., 2013). Height is up to 30 m but usually varies from 10 to 20 m (Londoño and Peterson, 1991). The juvenile bamboos usually reach the sunlit portion of canopy by 10 years of age, when they accelerate in growth (Smith and Nelson, 2011). They do not form

continuous pure stands, being mixed among the trees, yet achieve remarkable high densities ($2,309 \pm 1,149$ ind ha$^{-1}$) (Castro et al., 2013) and have significant ecological impacts. Thus, these forests support up to 40% less tree species diversity than nearby bamboo-free forests and from 30 to 50% less carbon stored as a consequence of the lower woody tree density (Silveira, 2001; Rockwell et al., 2014). Bamboo-dominated forests also have elevated tree mortality rates ($3.6 \pm 2.5$ % yr$^{-1}$) (Castro et al., 2013; Medeiros et al., 2013) when compared even to the typically fast-turnover forests in western Amazon (2.62 % yr$^{-1}$)

(Johnson et al., 2016). In the region, 74 different bamboo populations, that is, patches having individuals of the same internal age, have been so far identified, with a mean patch area of 330 km$^2$, and up to 2,570 km$^2$ for the largest patch (Carvalho et al., 2013). The mean lifetime of these bamboos was estimated in 28 years (Carvalho et al., 2013).

The locally synchronized death of semi-scandent bamboos produces large amounts of necromass in large patches over a short time. Decomposition of dead leaves and branches is rapid, but a layer of culms can remain intact on the forest floor for

up to three years (Silveira, 2001). When neighboring populations (patches) of bamboo go through reproductive events one after another in successive years, this is known in the literature as a flowering wave. The current hypotheses to explain this phenomenon include climatic variations; severe environmental pressures such as floods and fire (Franklin et al., 2010; Smith and Nelson, 2011); and incipient allochronic speciation – stochastically forming a small and rare temporally offset daughter patches at the margin of an expanding parent population. The individuals maintain (or synchronously amplify) their temporal

offset as they expand in order to maximize the once-in-a-lifetime chance of cross-pollination within the offset population (Carvalho et al., 2013).

Two main hypotheses, which are not competing but complementary, have been advanced to explain the dominance of semi-scandent bamboos in Amazon forests. Firstly, they cause elevated physical damage to trees by loading and crushing, while also suppressing recruitment of late succession tree species (Griscom and P. Mark S. Ashton, 2003). Secondly, they increase fire

probability via their mast seeding behavior followed by the synchronized death of the adult cohort, which produces large fuel





loads. The fire would then eliminate canopy trees, form gaps and inhibit tree recruitment, while creating an optimal environment for the bamboo seedling cohort. This latter hypothesis is called the 'bamboo-fire hypothesis' (Keeley and Bond, 1999). This hypothesis is attractive as it explains how bamboos can regain dominance of the forest after relinquishing space to trees when the adults die. Analysis of charcoal in soils of three Amazon bamboo-dominated forests sites showed a long history of fire

occurrence (McMichael et al., 2013). Smith and Nelson (2011) showed that fire disturbance favored the expansion of bamboos in the Amazon. Another study indicated that pre-Columbian people used fire and bamboo die-off patches to facilitate forest clearing and constructed geoglyphs, which, nowadays, can be found under the closed-canopy forest (Mcmichael et al., 2014). Although these studies do not support fire as the main driver of bamboo distribution ('bamboo-fire hypothesis'), they show associations between the bamboo die-off and increased fire occurrence, and potential human interactions on this processes.

Bamboo-dominated *terra firme* forests in the southwest Amazon can be detected in the optical bands of orbital sensors at the adult stage and the borders of each internally synchronized population can be detected after die-off events (Nelson, 1994). Carvalho et al. (2013) showed that the near infrared (NIR) band of the Thematic Mapper (TM)/Landsat-5 allowed the best discrimination between bamboo-free forest, forest with adult bamboo and forest with recently dead bamboo. Forests with adult bamboos showed higher reflectance in the NIR than bamboo-free or with recently dead bamboo. Forests in which the newly

sprouted cohort of seedlings is confined to the understory were not visually distinguishable from bamboo-free forest. The juvenile bamboo stays hidden in the understory up to 10 years of age, which is the moment they start reaching the canopy (Smith and Nelson, 2011; Carvalho et al., 2013). When analyzing Enhanced Vegetation Index (EVI) data from the Moderate Resolution Imaging Spectroradiometer (MODIS), processed by the Multi-Angle Implementation of Atmospheric Correction (MAIAC) algorithm (Lyapustin et al., 2012), Wagner et al. (2017) detected some patches of adult bamboo during a climate

driver study of Amazon forest greening. The bamboo patches presented two peaks of MODIS EVI per year (dry and wet seasons) compared to one peak observed in the wet season over bamboo-free forest.

Because the previous investigations used visual interpretation of satellite data and performed manual delineation of the bamboo areas, they were limited to the identification of large areas and constrained by the analyst's visual acuity. Further studies are therefore necessary to understand the bamboo life cycle, its spectral characteristics, as well as, to establish auto-

matic approaches for detecting die-off events in bamboo-dominated areas. These approaches can enable analyses of ecological processes associated with these events, such as the interactions between bamboo and fire (Keeley and Bond, 1999), bamboo flowering wave patterns (Franklin et al., 2010) and the distribution of 'bamboo-specialist' bird species (Kratter, 1997).

In this study, we developed and validated a method to map the die-off, spatial distribution and age structure of bamboo-dominated areas and investigated the relationship of bamboo with fire occurrence in the southwest Amazon. We also aimed

to provide near-term, spatially-resolved predictions of future bamboo behavior to allow our method to be further tested, val-idated, and improved over the coming years. Specifically, we evaluated the potential of MODIS (MAIAC) data (2000-2017) to automatically detect bamboo die-off events and live bamboo in the southwestern Amazon using a novel method based on bamboo life cycle characteristics. Using the die-off year as a time marker, we assessed the spectral variability of bamboo with cohort age, and applied this knowledge to predict the subsequent die-off year of each pixel from the distribution map for the



2017-2028 period. Using the complete map of bamboo populations' age structure and the active fire data product from MODIS, we analyzed the association between bamboo life cycle stage and fire frequency.

## 2 Material and Methods

### 2.1 Study area

5 The study area is located in the southwest Amazon between the longitudes 74º W and 67º W and latitudes 13º S and 6º S, covering parts of Brazil, Peru and Bolivia (Fig. 1). The predominant forest type is non-flooding open-canopy rain forest on terra firme, often dominated by bamboos of *Guadua* genera and mostly (93%) preserved from human disturbances (IBGE, 2006; Trancoso et al., 2010).

The most important soil types are chromic alisol, red-yellow argisoil, haplic cambisol, ferrocarbic podsol, haplic gleysol, 10 red-yellow latosol, chromic luvisol, and haplic plinthosol (dos Santos et al., 2011). In bamboo-dominated areas, the soils have a tendency to be more fertile, richer in exchangeable cations, more easily eroded, more poorly drained, and more clay-rich than the soils where bamboo is excluded (Carvalho et al., 2013). Naturally high erosion leads to a gently rolling hilly landscape (Mcmichael et al., 2014) with muddy streams and rivers. Based on a 19-year time-series of the Tropical Rainfall Measuring Mission (TRMM) satellite, annual rainfall ranges from 1800 mm to 3400 mm, with zero to five dry months (i.e., less than 100 15 mm mo$^{-1}$). The average temperature is 27 ºC. Minimum rainfall and temperature are in July (Dalagnol et al., 2017).

### 2.2 Satellite data and products

#### 2.2.1 MODIS (MAIAC) surface reflectance data

Daily surface reflectance data were acquired from the MODIS sensor, on board the Terra and Aqua satellites, from 2000 to 2017. They were corrected for atmospheric effects by the MAIAC algorithm (Lyapustin et al., 2012). The data were obtained 20 from the NASA Center for Climate Simulation (NCCS) repository (available at: ftp://dataportal.nccs.nasa.gov/DataRelease/). We used MAIAC surface reflectance and BRDF products at spatial resolution of 1 km, daily temporal resolution, in eight spectral bands: Red, 620-670 nm (B1); NIR-1, 841-876 nm (B2); Blue-1, 459-479 nm (B3); Green, 545-565 nm (B4); NIR-2, 1230-1250 nm (B5); Shortwave infrared-1 (SWIR-1), 1628-1652 nm (B6); SWIR-2, 2105-2155 nm (B7); and Blue-2, 405-420 nm (B8).

25 In order to minimize the differences in sun-sensor geometry between the MODIS scenes, the daily surface reflectance was normalized to a fixed nadir-view and a 45º solar zenith angle using a Bidirectional Reflectance Distribution Function (BRDF) and the Ross-Thick Li-Sparse (RTLS) model (Lucht and Lewis, 2000). Parameters of the RTLS model and BRDF kernel weights are part of the MAIAC product suite with temporal resolution of 8 days. Hence, the normalized surface reflectance, called Bidirectional Reflectance Factor ($BRF_n$) (Eq. 1), was calculated using the RTLS volumetric ($f_{vol}$) and geometric ($f_{geo}$) 30 parameters, and BRDF isotropic ($k_{iso}$), volumetric ($k_{vol}$) and geometric-optical ($k_{geo}$) kernel weights (Lyapustin et al., 2012).



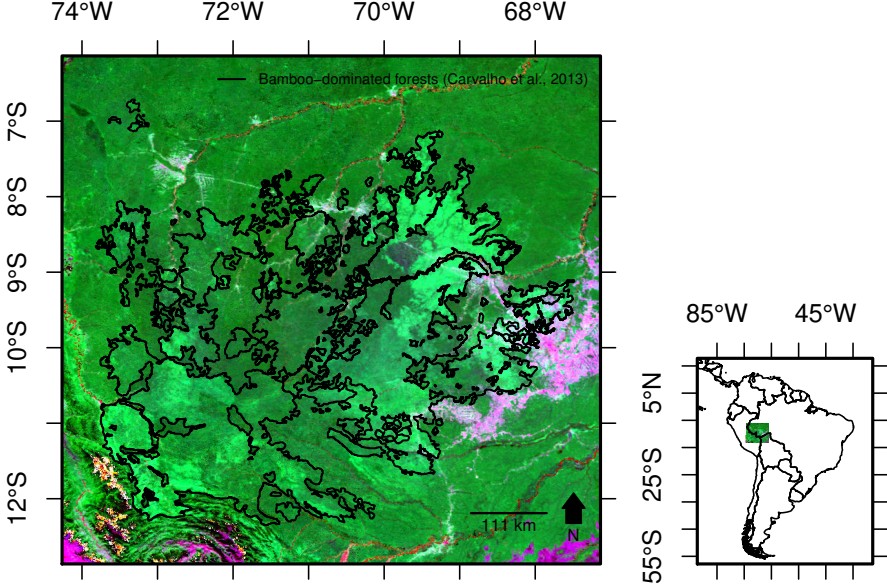

**Figure 1.** Bamboo-dominated forests in southwestern Amazon. The image at background is a false-color composite from MODIS (MAIAC) images of bands 1 (Red), 2 (NIR) and 6 (shortwave infrared), in RGB, respectively, in August 2015. The black lines indicate the perimeter of the bamboo-dominated areas delineated in a previous study (Carvalho et al., 2013).

$$BRF_n = BRF \times \frac{k_{iso} - 0.04578 \times k_{vol} - 1.10003 \times k_{geo}}{k_{iso} + f_{vol} \times k_{vol} + f_{geo} \times k_{geo}} \tag{1}$$

The $BRF_n$ data were aggregated into 16-day composite intervals by calculating the median on a per-pixel basis. The composites were then merged and converted to geographic projection (datum WGS-84). All these procedures were implemented in R language (R Core Team, 2016).

5     Annual composites of MODIS NIR surface reflectance data were selected between July and September to minimize cloud coverage. Furthermore, during these months, the bamboo patches at the adult stage present a well-defined phenological response (peak in MODIS EVI), which is not present in primary forests without bamboo dominance (Wagner et al., 2017). When useful data were not available in the time series due to cloud cover or bad pixel quality retrievals, an imputation method was applied to fill the gaps using the whole time series. As the bamboo-dominated forests present a seasonal spectral response,

10   the imputation was conducted by the Seasonal and Trend decomposition using Loess (STL) method (Cleveland et al., 1990). This method decomposes the signal into trend, seasonal and irregular components, interpolates the missing values, and then reverts the time series. It is effective when dealing with missing values in seasonal signals when compared to other imputation methods (Steffen, 2015).




### 2.2.2 TM/Landsat-5 surface reflectance product

An annual time series of Thematic Mapper (TM)/Landsat-5 data was obtained from 1985 to 2000 in order to visually detect die-off events that occurred in the last life cycle of bamboo and validate age predictions. We selected atmospherically corrected surface reflectance images (Landsat collection 1 Level-1) (available at: https://earthexplorer.usgs.gov/) from the quarter July-

August-September to increase the chances of obtaining cloud-free data and reduce spectral variations associated with vegetation seasonality.

### 2.2.3 Tree cover product

In order to mask areas that were not covered by intact forests (deforested, degraded, and secondary forests, pastures and swidden fields) and to analyze the tree cover of the bamboo-dominated forests, we used the global forest cover loss 2000–2016

dataset (Available at: https://earthenginepartners.appspot.com/science-2013-global-forest/download_v1.4.html). The dataset is based on Landsat time series data at 30 m spatial resolution (Hansen et al., 2013), and consists of tree cover percentage, gain, and loss during 2000-2016, and a mask indicating permanent water bodies. It was re-sampled to 1 km spatial resolution using the average interpolation in order to match the resolution of the MODIS (MAIAC) data. A mask of intact forests was created using the tree cover data to select pixels: (i) without permanent water bodies, (ii) without gain or loss of tree cover during the

2000–2016 period; and (iii) above a threshold of 95% tree cover to detect and filter out non-forest pixels. In order to create a yearly non-forest fraction mask, the number of loss pixels (30 m) of each year from 2001 to 2016 were counted inside the MODIS cells (1000 m), and then accumulated from 2000 to 2016. The forest and non-forest areas in 2000 were considered as above and below a threshold of 80% tree cover, respectively.

The bamboo-dominated forests area delineated by Carvalho et al. (2013) was used as reference to assess the tree cover

variability in forests with and without bamboos. This map was obtained in the previous study by visual interpretation of live-adult bamboo using two Landsat mosaics 10 years apart from each other (1990 and 2000), supported by the known locations and dates of five bamboo dominated areas. Tree cover percentiles (0.01, 0.5, and 0.99) were calculated considering the pixels inside the bamboo-dominated areas.

Because previous studies showed that the NIR spectral interval was able to distinguish forests with and without bamboos

on the canopy (e.g., Carvalho et al., 2013), we inspected the MODIS NIR-1 reflectance for pixels with tree cover below the 1st percentile, between 1st and 99th percentile, and above the 99th percentile. The normal distribution of the pixel population was assessed using a two-sided Kolmogorov-Smirnov test at a 1% significance level. For normal distribution, the average and standard deviation of distributions were computed. For skewed distributions, the average, standard deviation and skewness parameter (xi) were estimated (Fernandez and Steel, 1998).





### 2.2.4  MODIS active fire detections product

MODIS active fire data from Aqua satellite at 1 km spatial resolution was obtained from the Brazilian Institute of Space Research (INPE) Burn Database (Available at: http://www.inpe.br/queimadas/bdqueimadas/) for the period of 2002–2017 over the study area. This dataset corresponds to geolocations of active burning areas in the moment of satellite overpass.

5  ### 2.3  Bamboo life cycle spectral characteristics

### 2.3.1  Die-off detection and validation

For bamboo die-off detection, we assumed a fixed bamboo lifetime of 28 years - based on Carvalho et al. (2013) findings, and assumed the signal coming from the trees as constant over time. Therefore, inter-annual reflectance variations were attributed to structural changes in the canopy related to bamboos. We tested two different NIR bands of MODIS: NIR-1 band 2 (841-876 10 nm) and NIR-2 band 5 (1230-1250 nm). Both bands are sensitive to canopy structure scattering, but NIR-2 is also partially sensitive to leaf/canopy water scattering (Gao, 1996). The automatic detection of bamboo die-off was conducted by assessing the point of maximum correlation between each pixel's MODIS (MAIAC) NIR reflectance time series and a bilinear model following our hypothesis: a linearly increasing NIR reflectance vector from 1 to 28 years (Y = x) followed by an abrupt reflectance decrease at 29 years of bamboo age (die-off event; Y = 0). The Pearson's correlation coefficient (r) between the 15 NIR reflectance time series and the bilinear model for a given pixel was iteratively tested by shifting the position of the NIR time series inside the bilinear model vector. The position showing the highest r corresponded to the estimated age of that pixel from which the die-off year was retrieved. Only pixels with correlations very significant (p < 0.001) were selected.

For validation purposes, we compared the automatically detected die-off events with recently dead bamboo areas visually detected in false color composites obtained from MODIS bands 1, 2 and 6 in RGB, respectively. In this color composite 20 (Fig. 1), adult bamboo patches show bright green color due to the comparatively higher NIR reflectance, while dead bamboo patches present dark blue/gray color. The visual inspection of bamboo die-off using MODIS and Landsat data was consistent with five bamboo mass flowering events observed in the field (Carvalho et al., 2013). In each of the dead bamboo patches visually detected, the geographic location and die-off year were registered for a sample of 5 pixels. A total of 390 geolocations with corresponding year of bamboo death was obtained over 78 dead bamboo patches for the 2001–2017 period. The cross- 25 validation consisted in calculating the exact die-off year detection's accuracy, Pearson's correlation and p-value, and the root mean square error (RMSE) between the automatically detected and visually interpreted die-off year.

### 2.3.2  Spatial distribution detection

In order to map the spatial distribution of bamboo-dominated forests for the whole area, we first detect the live bamboo and then merge this map with the die-off detection map (2001-2017). We used two assumptions to map the live bamboo: (i) over 30 the 18 years' period, it should present mean NIR reflectance equal to or greater than the median signal of bamboo-free forests; and (ii) it should present an increasing NIR reflectance over time. The median bamboo-free forest signal was derived using the





tree cover mask and a threshold that excluded all the potential bamboo-dominated pixels. The threshold was defined as the tree cover percentage above the 99th percentile from bamboo-dominated forests as delineated by Carvalho et al. (2013). We tested whether the mean NIR reflectance of each bamboo-dominated pixel was statistically lower than the forest median signal using the Student's t-test, and excluded those pixels. Furthermore, we obtained a linear regression model between the reflectance

of each pixel in the time series and a linear increasing vector to identify reflectance increase over time in the bamboo areas. We selected only pixels that showed a very significant ($p < 0.001$) and positive regression slope, indicating the reflectance increase in the NIR. In order to assess the consistency of the map, we compared it with the available bamboo-dominated forests distribution map from Carvalho et al. (2013).

### 2.3.3 Bamboo cohort age and spectral variability

We used the die-off map to analyze the variation of bamboos spectral response with age. Data from all MODIS bands were extracted using the estimated die-off year with very significant correlation ($p < 0.001$) as a starting point. Bamboo cohort age was then calculated backwards and forwards in time during the 2000-2017 period. Reflectance percentiles (0.01, 0.5, 0.99) per age were calculated obtaining, what we called, empirical bamboo-age reflectance curves.

The spectral variability with cohort age was also analyzed in relation to bamboo-free signal in order to assess the separability

of forests with and without bamboo. Pearson's correlation between the median bamboo-free signal, as obtained in a previous section, and bamboo-dominated forest pixel's signal were calculated and assessed as a function of cohort age. The assessment was conducted using the NIR-1 and NIR-2 bands.

### 2.3.4 Die-off prediction

The bamboo cohort age and the reflectance of the MODIS NIR-1 and NIR-2 were then applied not only for detection of die-off

from 2000-2017, but also for prediction of the die-off year in the 2018–2028 period. Annual TM/Landsat-5 color composites (bands 2, 4 and 1 in RGB) were visually inspected to identify bamboo die-off events that occurred during the 1985-2000 period. We assumed that the die-off events that happened in this period would happen again in the next life cycle of the bamboo, up to 2028. Therefore, we added 29 years to the visually detected die-off year in order to match the next life cycle. A total of 175 geolocations with corresponding years of death were collected in 35 dead bamboo patches. A cross-validation was conducted

by calculating the same metrics as in the die-off detection section. The distribution of predictions was tested for normality using a two-sided Kolmogorov-Smirnov test at a 1% significance level. The size distribution of all bamboo populations was assessed by quantifying the number, minimum, maximum, mean and median size of bamboo patches. In order to filter out noise in the predictions (i.e. loose pixels), the minimum patch size was set to 10 km$^2$.

### 2.4 Relationship between bamboo die-off events and MODIS active fire detections

Active fire detections from MODIS/Aqua during 2002-2017 were filtered using the yearly non-forest fraction mask considering a threshold of 0% non-forest pixels. This assured that active fires occurring over deforested and degraded forests, pastures or



swidden areas were removed, and only pixels over forested areas remained in each year. The active fires were plotted over the bamboo spatial distribution map in order to visualize where the fire occurred. The number of fires occurring over live bamboo and dead bamboo (died-off during 2001-2017) was calculated.

In order to test whether there is a higher fire occurrence over recently dead bamboo than live bamboo, the active fire detections were analyzed as a function of dead (28, 0 and 1 years) and live bamboo (2 to 27 years) classes. For this purpose, each active fire detection was labeled accordingly to the bamboo age of the pixel where it occurred and then merged into the two classes. We controlled for three factors that can affect fire probability: area of bamboo mortality, climate, and proximity to ignition sources. Since the total area of a specific age class could interfere with fire frequency, that is, more area would mean higher probability of fire occurrence, we normalized the fire frequency by the area (ha) of its respective age class within the buffer with most fire occurrences, in the year of fire occurrence. Severe droughts affected Amazonia in 2005, 2010 and 2015/2016, and especially the southwest in 2005 (Aragão et al., 2007; Phillips et al., 2009; Lewis et al., 2011; Aragão et al., 2018). As drought years can enhance fire occurrence in Amazonia (e.g., Brando et al., 2014; Aragão et al., 2018), we analyzed separately the fire frequency in regular and drought years. In order to assess the influence of ignition sources on the fire occurrence, we filtered active fire detections using buffers of 1, 2 and 3 km around the non-forested areas using the yearly non-forest fraction mask and assessed the number of active fire detections considering each buffer.

The area-normalized fire frequency over dead and live bamboo was compared using a two-way Analysis of Variance (ANOVA). We tested the effects of bamboo life stage (live or dead), year of fire occurrence and their interactions over active fire detections.

## 3 Results

### 3.1 Tree cover analysis

Bamboo-dominated forest as mapped by Carvalho et al. (2013) spanned a very narrow range of values in the Landsat-derived percent of tree cover product. The 1st and 99th percentiles of tree cover in the bamboo areas were 96.95% and 99.88%, respectively (Fig. 2). The median was 99.18%. Forests identified as bamboo-free by the 2013 study had tree cover above the 99th percentile at the northeast of the study area, but below the 1st percentile at the southwest of the study area. At the northwest, bamboo-free forests presented tree cover similar to that of bamboo-dominated, i.e., between the 1st and 99th percentile.

The MODIS NIR-1 reflectance values over the 2000-2017 period in bamboo-free forests that had tree cover above the 99th percentile of bamboo-dominated areas followed a normal distribution (p = 0.33) and showed the lowest standard deviation (SD) compared to the bamboo-dominated forests (mean = 27.3% reflectance; SD = 0.9%) (Fig. 3). Bamboo-free forests that had tree cover below the 1st percentile of bamboo-dominated areas presented a left-skewed distribution with similar reflectance to the 99th percentile but with higher SD (mean = 27.2%, SD = 2.6%, and xi = 1.2). Bamboo-dominated forests (pixels inside the hatched polygon in Fig. 2) presented a right-skewed distribution with higher NIR-1 reflectance than the bamboo-free forests (mean = 28.7%, SD = 2.1% and xi = 1.9).



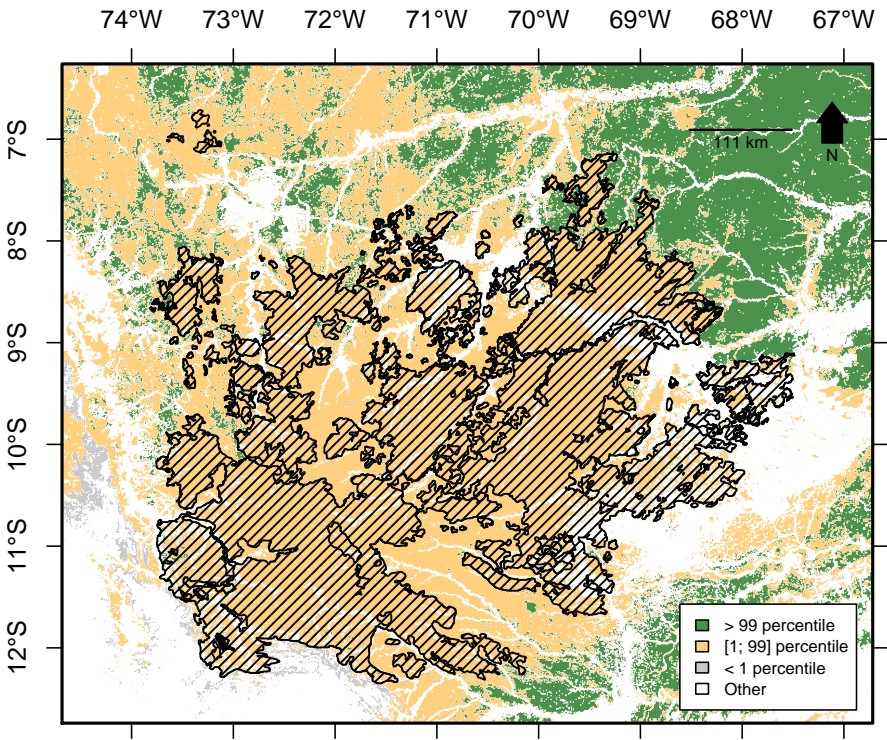

**Figure 2.** Spatial distribution of stable tree cover percentage percentiles (filtered for tree cover gain and loss, and for water bodies), indicating pixels below, above and within the 1st to 99th percentile range of tree cover found in bamboo-dominated forest (hatched), as delineated by Carvalho et al. (2013).

## 3.2 Bamboo life cycle spectral characteristics

### 3.2.1 Die-off detection

When we applied our automatic die-off approach over the canopy-scattering (NIR-1 band 2) and canopy-water (NIR-2 band 5) sensitive MODIS NIR bands, differences in detected bamboo areas were observed (81480 km$^2$ for NIR-1 and 86628 km$^2$

5 for NIR-2). Despite these differences, the resultant die-off year maps were consistent to each other (Figs. 4A and 4B) with 81% of the detected die-off events located inside the bamboo-dominated area, as reported by Carvalho et al. (2013). The die-off patches that were detected over a 18 years period inside the previous bamboo-dominated forest map represented 40.7% and 42.7% of the total bamboo area using MODIS NIR-1 and NIR-2, respectively. In Figures 4A and 4B, 83.6% of the dead bamboo pixels generated from the two NIR bands showed the same year of death between the maps. Interestingly, some small

10 patches between 8-9ºS and 73-74ºW presented a unidirectional wave of mortality from north to south with a delay of one year between adjacent patches.




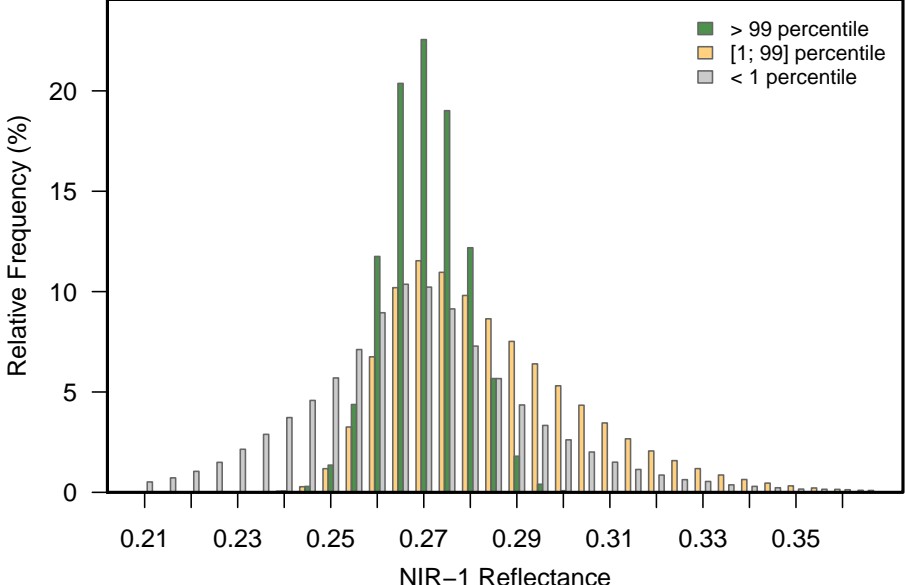

**Figure 3.** Relative frequency of MODIS NIR-1 reflectance (band 2) from pixels with tree cover percentage below, above and within the 1st to 99th percentile range of tree cover found in bamboo-dominated forest (hatched), as delineated by Carvalho et al. (2013).

The correlation coefficients found in the mapped pixels with significant relationship with our bilinear model (p < 0.001) were very strong (r > 0.7). When the automatic die-off estimates were cross-validated with the visually inspected die-off from 2001–2017, the accuracy from NIR-2 was slightly higher (82.6%) than that from NIR-1 (79.3%) (Figs. 4C and 4D). Both bands showed similarly strong Pearson's correlation (r > 0.99, p < 0.01), whilst NIR-1 showed slightly lower RMSE (0.48 years) than that from NIR-1 (0.54 years).

### 3.2.2 Spatial distribution of bamboo-dominated forests

The bamboo-dominated forests were mapped by merging the die-off detection during 2001–2017 with the live bamboo detection (Fig. 5). The die-off detection was based on both MODIS NIR-1 and NIR-2, which presented high accuracies and mapped slightly different bamboo patches in Figure 4. The live bamboo detection was based only on NIR-1, which did not saturate with bamboo growth over time in Figure 6. A total of 155,159 km$^2$ of bamboo-dominated forest was detected in the area. Of these, 112,570 km$^2$ or 72.5% were located inside the bamboo forest mapped by Carvalho et al. (2013). A total of 68.8% of the bamboo forest area from Carvalho et al. (2013) was covered by the detection. A few large patches were found outside of the previously mapped bamboo spatial distribution, such as in 11.5º S; 70º W, and 13º S; 71º W.



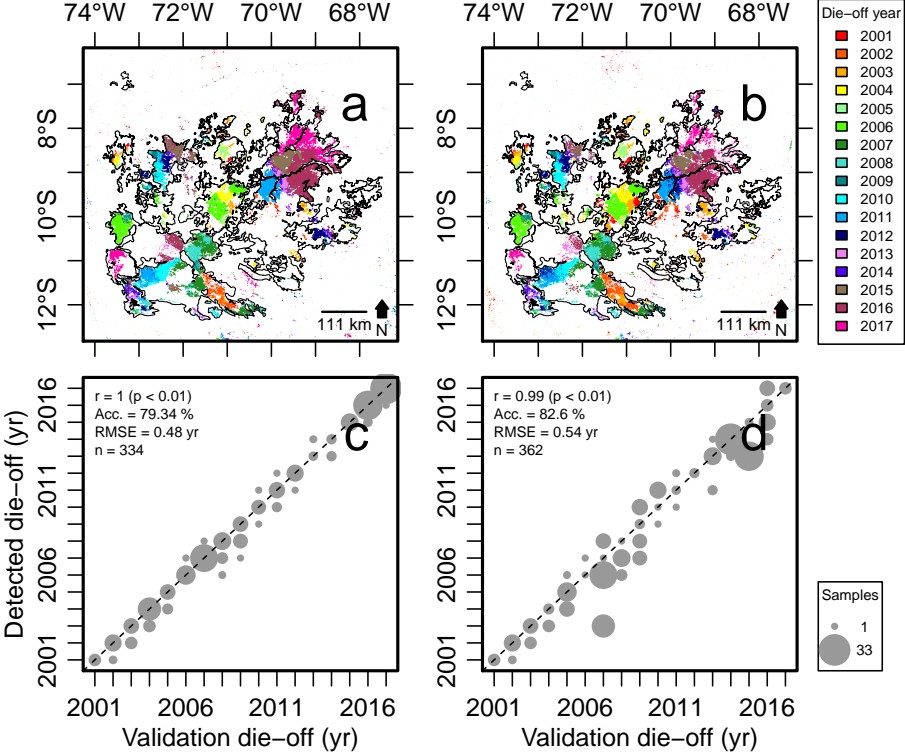

**Figure 4.** MODIS bamboo die-off detection map from 2001 to 2017 using the bilinear model of expected near infrared (NIR) reflectance variations as a function of bamboo cohort age, for (a) NIR-1 and (b) NIR-2. Cross-validation between detected die-off and visual interpreted die-off on MODIS false-color composites (2000-2017) for (c) NIR-1 and (d) NIR-2. The dashed line represents the 1:1 line.

### 3.2.3  Bamboo cohort age and spectral variability

The reflectance of the MODIS NIR-2 and the two SWIR bands slowly increased with bamboo development up to about 12 years of age, and then increased very steeply from 12-14 years (Fig. 6). NIR-1 did not show the same reflectance increase up to 12 years as NIR-2, but also showed the steep increase in reflectance between 12-14 years. A pronounced but temporary

5  dip in Red and Blue-2 reflectance occurred concurrently with this brief and rapid NIR and SWIR increase. Green reflectance increased up to about 17 years then leveled off. The response of two SWIR bands and the NIR-2 band all leveled off after 15 years. The NIR-1, however, showed increasing reflectance over the cohort remaining life span, until the age of synchronous die-off. The bamboo die-off was marked by a sharp decrease in MODIS NIR-1 and NIR-2 reflectance between 28 and 29 years of age (Fig. 6). A reflectance change with bamboo death was not well defined in the SWIR-1 and SWIR-2 bands. The

10  reflectance of all bands presented high dispersion with coefficients of variation ranging from 5.9 to 20.3%.

The mean Pearson's correlation between the median bamboo-free forest and bamboo-dominated forests NIR-1 reflectance decreased from 0.41 to -0.02 in the transition from juvenile (1-14 years) to adult bamboo stage (15-28 years) (Fig. 7, black boxes). The correlation in the partially water-sensitive NIR-2 did not follow the same pattern (Fig. 7, orange boxes). In NIR-2,



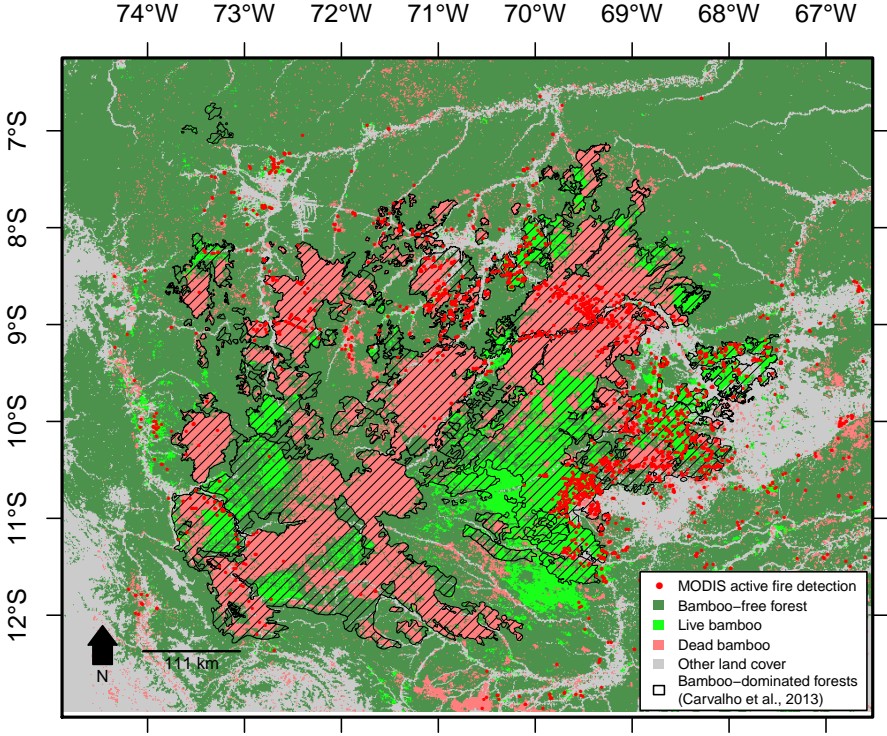

**Figure 5.** Bamboo-dominated forests map and MODIS active fire detections during 2002–2017 (red dots). Orange pixels are bamboo die-off patches detected during 2001-2017 using the bilinear model. Light green pixels are bamboo that did not die-off, but showed increasing NIR signal during 2001–2017, and presented greater NIR mean than forest. Dark green pixels are bamboo-free forests. Gray pixels are other land cover classes. The hatched polygon represents the bamboo-dominated forests delineated by Carvalho et al. (2013).

the correlation was similar in juvenile (r = 0.19) and adult bamboo stages (r = 0.2). The correlation's standard deviation was 0.14 and 0.2 for juvenile and adult stages in both bands.

### 3.2.4 Die-off prediction

Based on the NIR-1 and NIR-2 reflectance from 0–28 years of age, we predicted the die-off year from 2000 to 2028 for each
5   patch (Fig. 8A and B, respectively). The estimated die-off years using the empirical curves during 2001-2017 were 85% similar to the detection using the initial bilinear model (Fig. 4). The empirical curves achieved an accuracy of 75.45% (RMSE = 1.11 years) and 69.23% (RMSE = 1.08 years) for NIR-1 and NIR-2, respectively, on predicting the exact die-off year from 2001-2017, when compared to the visual inspection of MODIS color composites. Die-off prediction during 2018–2028 using the empirical curves (Fig. 6) with NIR-1 and NIR-2 were inspected for consistency using the visual interpretation of TM/Landsat-5
10   time series (Fig. 8C and 8D, respectively). NIR-1 and NIR-2 presented low accuracy (20.5 and 3%, respectively) to predict the




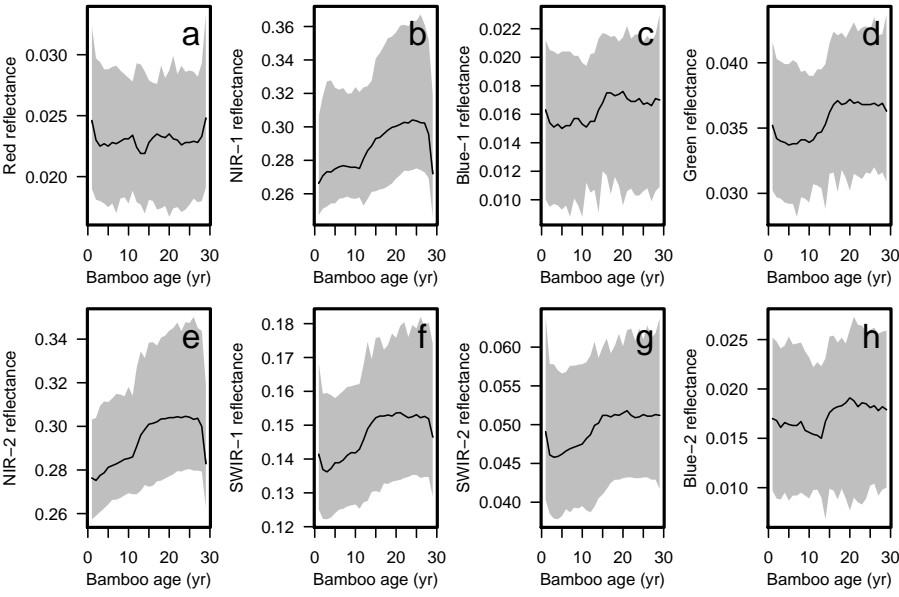

**Figure 6.** Empirical bamboo-age reflectance curves at ages 0-28 years from MODIS bands 1 to 8 (a)-(h). Black lines represent the median, while the shaded gray areas represent the 1st and 99th percentile.

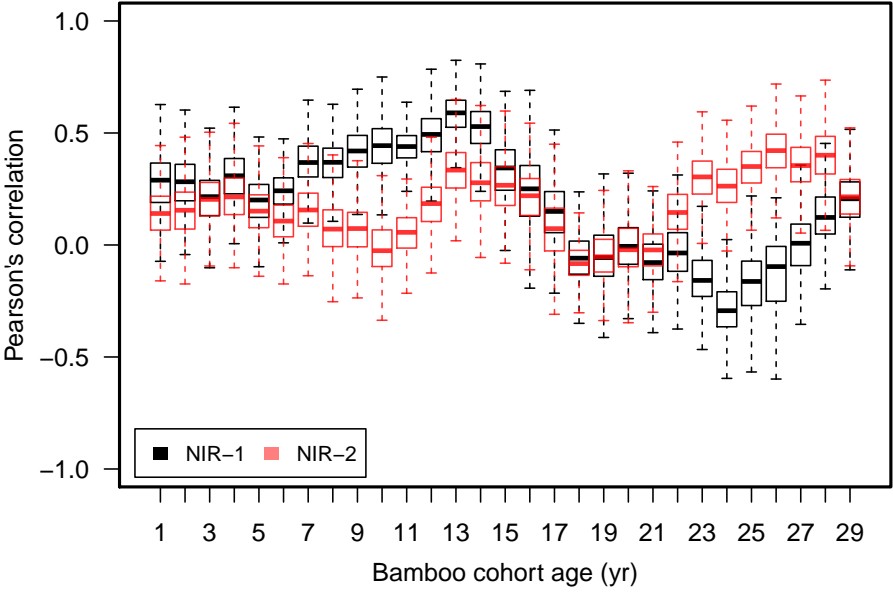

**Figure 7.** Pearson's correlation coefficients between the median reflectance of bamboo-free forest with the pixel spectral response of bamboo-dominated forests. The results are plotted as function of the bamboo cohort age for MODIS NIR-1 (in black) and NIR-2 (in orange).



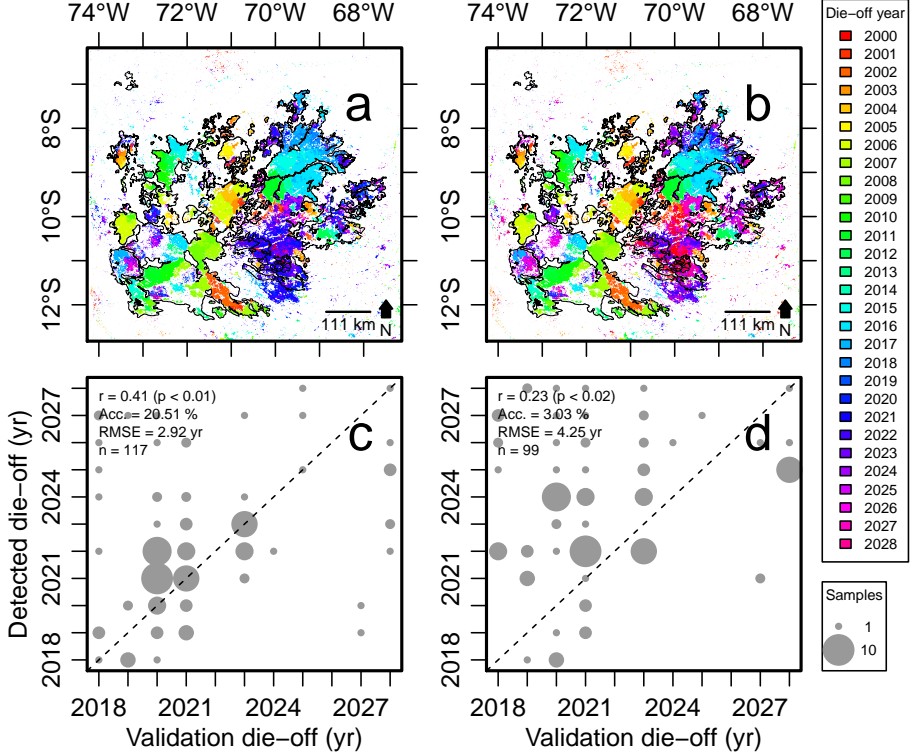

**Figure 8.** MODIS bamboo die-off prediction map from 2000 to 2028 using the empirical curves of the near infrared (NIR) reflectance as a function of bamboo cohort age, for the (a) NIR-1 and (b) NIR-2. Cross-validation between predicted die-off and visual interpreted die-off from previous life cycle in Landsat false-color composites (1985-2000) are shown for the (c) NIR-1 and (d) NIR-2. The dashed line represents the 1:1 line.

exact die-off year with, high RMSE (2.92 and 4.25 years, respectively) and significant weak to moderate correlations (r = 0.41 and p < 0.01; 0.23 and p < 0.02, respectively).

The residuals distributions of both NIR-1 and NIR-2 prediction models (Fig. 9A and 9B, respectively) were not significantly different from normal (p > 0.1). The NIR-1 model had a mean age error closer to zero (-0.7 years) than that observed from 5 NIR-2 (-1 years). This indicates an average underestimate of the true die-off year when using MODIS NIR-1 and NIR-2, respectively. The standard deviation of the age model residuals was smaller for NIR-1 (5 years) than for NIR-2 (9 years).

As the MODIS NIR-1 prediction model (Fig. 8C, Fig. 9A) showed higher precision and less bias than the model based on NIR-2 (Fig. 8D, Fig. 9B), we extracted the predicted die-off years from the NIR-1 model to estimate the total area of bamboo die-off per year (Fig. 10) and bamboo population (patch) size distribution (Table 1). Total die-off per year was different from a 10 uniform temporal distribution (p < 0.01). For a uniform distribution, the annual die-off areas would be close to the average of 5350 km$^2$. Within the period 2000–2017, the years 2006, 2007, 2011, 2015 and 2016 showed higher than average die-off area



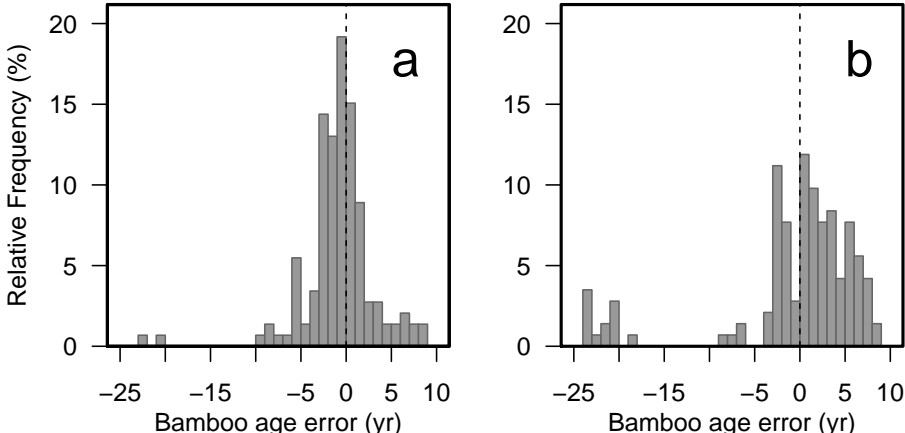

**Figure 9.** Residual distributions of the prediction models between 2018–2028 for MODIS (a) NIR-1 and (b) NIR-2. The dashed line represents age residual = 0.

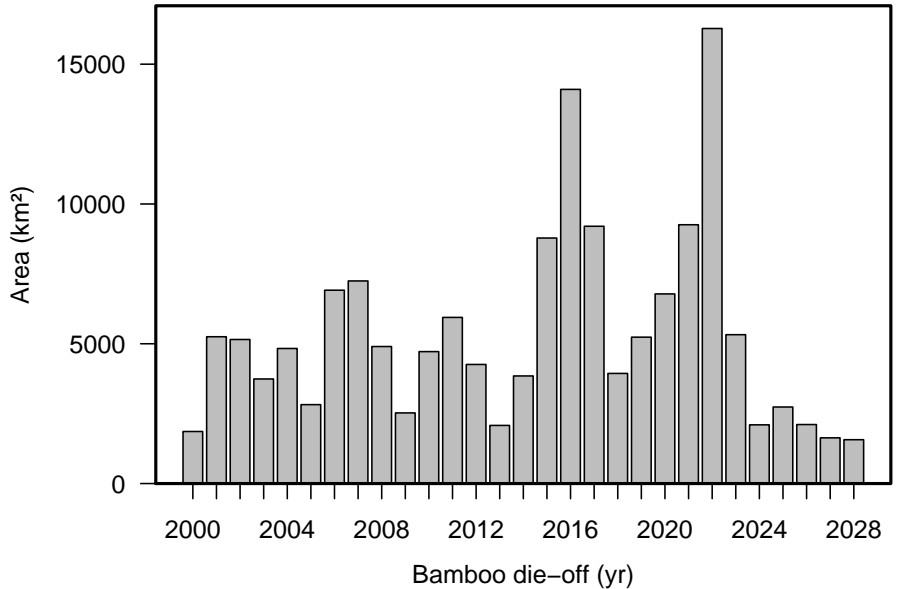

**Figure 10.** Distribution of predicted bamboo die-off area per year between 2000 and 2028 from MODIS NIR-1.

(Fig. 10). The largest die-off area was observed in 2016 (14099 km$^2$). For the 2018–2028 predicted period, the year of 2022 is expected to show the largest bamboo die-off area (16276 km$^2$).

The detection for 2001–2008, a period that matches the time interval analyzed visually by Carvalho et al. (2013), showed 372 die-off patches with mean size of 80 km$^2$ and maximum size of 2234 km$^2$ (Table 1). Carvalho et al. (2013) found 74 patches with mean size of 330 km$^2$ and maximum size of 2570 km$^2$ during the same period. The detection for 2001–2017





**Table 1.** Bamboo patch sizes obtained from die-off prediction using MODIS NIR-1 filtered by a minimum patch area of 10 km$^2$, and comparison of results with those from Carvalho et al. (2013).

| Study | Period | n | Mean (SD) | Range (min, max) | Median |
|---|---|---|---|---|---|
| Carvalho et al. (2013) | 2001–2008 | 74 | 330 (?) | ?, 2570 | ? |
| This study | 2001–2008 | 372 | 79.56 (242.89) | 10, 2234 | 21 |
| This study | 2001–2017 | 802 | 84.57 (310.39) | 10, 6162 | 20 |
| This study | 2018–2028 | 778 | 33.84 (72.38) | 10, 1154 | 17 |
| This study | 2000–2028 | 1603 | 59.05 (226.66) | 10, 6162 | 18 |

showed 802 patches with mean size of 85 km² and maximum size of 6162 km² (Table 1). Some patch structures had long and linear perimeters, while others had rectangular shapes (for example near 69º 45' W, 8º 48' S, and 71º 13' W, 9º 47' S) or rounded borders (for example near 70º 45' W, 9º 39' S). We also detected a unidirectional flowering wave from north to south in the patch between 8-9º S and 73-74º W, which was also reported by Carvalho et al. (2013).

## 3.3 Relationship between bamboo die-off events and MODIS active fire detections

Active fire detections were not found in all bamboo patches that died (Fig. 5). We found a total of 2371 MODIS active fire detections inside bamboo-dominated forests between 2002 and 2017, from which 1424 detections (60%) occurred in bamboo patches that died-off and 947 detections (40%) occurred in live bamboo patches. Active fires were detected mostly near non-forested areas (Fig. 5 in gray). When we excluded the detections up to 1, 2 and 3 km around these areas, the total detections

decreased to 1330 (56%), 18 (0.76%) and 3 (0.12%), respectively.

In overall, there was a similar number of active fire detections per hectare in dead and live bamboo (0.18 fires ha$^{-1}$) (Fig. 11). The ANOVA did not show statistically significant differences in the area-normalized mean active fire detections for the interaction between bamboo stage (dead or live) and year of fire occurrence factors (p = 0.67). Individually, bamboo stage also did not show statistical significance on area-normalized mean active fire detections (p = 0.986). On the other hand, year of fire

did show (p < 0.01). The years 2017 and 2016 presented significant higher area-normalized mean active fire detections (0.46 and 0.35 fires ha$^{-1}$, respectively) than the other years (p < 0.01).

For severe drought years, the area-normalized active fire detections in 2005 (0.32 and 0.18 fires ha$^{-1}$), 2010 (0.22 and 0.12 fires ha$^{-1}$), 2015 (0.35 and 0.20 fires ha$^{-1}$) and 2016 (0.57 and 0.33 fires ha$^{-1}$) over dead and live bamboo, respectively, were not statistically different than the mean fire frequency in dead and live bamboo (0.18 fires ha$^{-1}$; p > 0.1). Even though, drought

years presented in average 45% higher area-normalized mean active fire detections in dead (0.342 fires ha$^{-1}$) than live (0.236 fires ha$^{-1}$) bamboo.





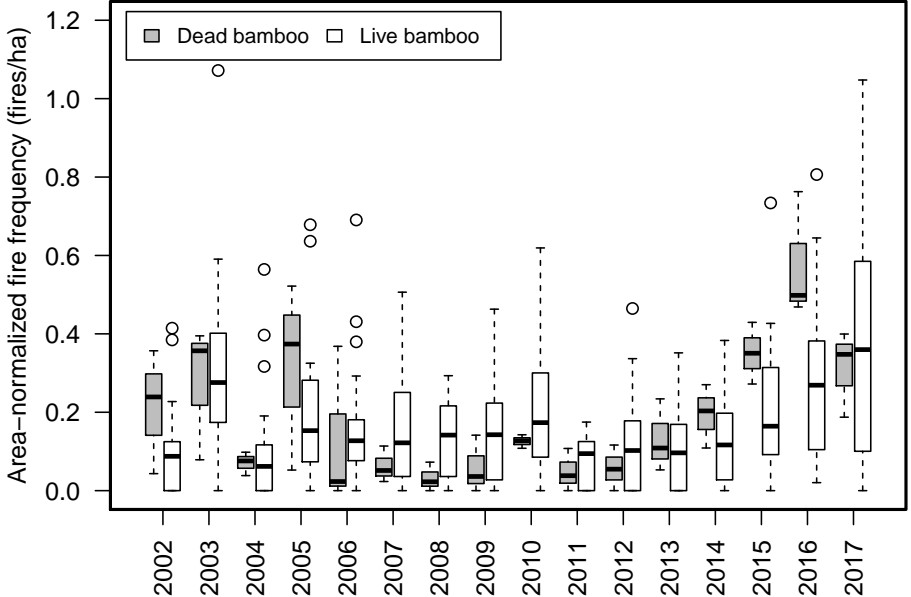

**Figure 11.** Area-normalized MODIS fire frequency during 2002–2017. Gray boxes represent fire in dead bamboo (28, 0 and 1 years) and white boxes represent fire in live bamboo (2 to 27 years).

## 4   Discussion

### 4.1   Tree cover of bamboo-dominated and nearby forests

We found that the tree cover of bamboo-dominated forests had a narrow range of values (96.95 to 99.89%) and was below the tree cover values of the closed forests nearby (above 99.89%). This suggests that these forests have a largely-closed canopy

5   but are slightly more open than closed forests without bamboo. Evergreen trees are the dominant life form over most of the southwest Amazon forests, including the ones that bamboo is very abundant. The trees generally comprise 50% or more of the canopy area in a Landsat or MODIS pixel, even when the bamboo cohorts are at adult stage and dense (Carvalho et al., 2013). They also fully dominate the canopy during 30% of the bamboo life cycle, while juvenile bamboo is confined to the forest understory (Smith and Nelson, 2011). The presence of canopy trees could explain why the tree cover is so high. Even though,

10   the tree cover percent of bamboo-dominated forests was slightly smaller than the bamboo-free areas. We believe this might be related to (i) an increased gap opening associated with faster forest dynamics and tree mortality of these areas influenced by bamboo (Castro et al., 2013; Medeiros et al., 2013), or (ii) artifacts of the tree cover computation method that uses the pixels' reflectance from Hansen et al. (2013).

The MODIS NIR-1 reflectance was normally distributed over bamboo-free forests, while it showed a right skewed distribu-

15   tion over the bamboo-dominated forests (Fig. 3). This result was expected considering that old-growth forests are more or less



stable over time, and the NIR reflectance increases in bamboo-dominated forests when they reach the canopy after 12 years of age (Smith and Nelson, 2011).

## 4.2 Automatic detection of bamboo die-off

The automatic detection of bamboo die-off performed very well with an accuracy above 79% when estimating the exact year

of bamboo death and a mean error of 0.5 years. When comparing the NIR-1 and NIR-2 bands, the leaf/canopy water sensitivity from NIR-2 might have contributed for a better performance on bamboo-die off detection. The die-off detection conducted in previous works was solely based on the visual inspection of Landsat and MODIS color composites, thus leading towards the identification of big clusters of pixels that went through die-off (Table 1) (e.g., Nelson, 1994; Carvalho et al., 2013). Our method is automatic, easy to implement, and can detect relatively small patches because it runs on a per pixel basis. However,

we do not advise to attempting detection of very small patches (e.g. $< 10 \text{ km}^2$) when using MODIS data due to limitations of the spatial resolution of the sensor (1 km). It is important to note that the detected bamboo die-off areas were not confounded with recently deforested areas, as the tree cover product did not point out forest losses in bamboo die-off areas. Since the method can detect bamboo die-off without a priori knowledge of the bamboo spatial distribution (Fig. 5), it could be used to better describe and understand the spatial organization of the bamboo stands that show synchronized die-off in forests around

the world.

    The high detection accuracy of bamboo die-off events also highlights the quality of the MODIS (MAIAC) data, which are suitable for bamboo-dominated forests mapping. The MAIAC algorithm improves the accuracy of cloud detection, aerosol retrieval and atmospheric correction compared to the standard MODIS product processing (Hilker et al., 2014). Combined with the appropriate normalization for sun-sensor-target geometry using BRDF modeling, the MAIAC contributed to minimize

inter-annual artifacts in the time series for an accurate detection.

## 4.3 Spatial distribution of bamboo-dominated forests

A total of 155,159 km$^2$ (15.5 million ha) of bamboo-dominated forests were mapped in the southwestern Amazon by combining the automatic detection of die-off and live bamboo (Fig. 5). Most of the detected areas (72.5%) were located inside the 16.5 million ha of the bamboo-dominated forests mapped by Carvalho et al. (2013), although covering only 68.8% of the

previous detected areas. This difference was part due to the increased land cover change in the region past-2010 – period which Carvalho et al. (2013) performed its analysis, areas which our method did not detect as bamboo-dominated forests. Despite the differences, we detected clusters of pixels that are very likely bamboo-dominated patches outside of the previously mapped areas (e.g., 11.5 ºS, 70 ºW, and 13 ºS, 71 ºW). These areas should be further investigated in field to verify if they are indeed bamboo-dominated forests.

Compared to our results with 1 km spatial resolution, the map from Carvalho et al. (2013) (30 m spatial resolution) under-estimated the bamboo-dominated forests in the order of 30%. A possible explanation is that the authors considered live-adult bamboo and used only two Landsat mosaic images 10 years apart from each other (1990 and 2000) for mapping, thus not observing part of the bamboos that were at juvenile stage and hidden in the understory at that time. Another possibility is the





limitation of visual interpretation and manual delineation of small bamboo patches. Our map was obtained on a per-pixel basis by assessing each pixel's spectral trajectory, thus reducing errors of omission by considering both live and dead bamboo for mapping, and by using a longer time series (18 years) for the detection.

The potential limitations of our map include the coarser spatial resolution (1 km) when compared to the previous map
(30 m). In addition, we likely underestimated the true bamboo distribution because of the previously discussed uncertainties in detecting juvenile bamboos, given the limited temporal coverage of the MODIS (MAIAC) time series. A more accurate mapping of bamboo spatial distribution would require that all bamboo died-off during the time series (i.e. requiring at least 28 years of data). At this moment, the only dataset that have such temporal coverage would be that from the Landsat satellites, which have a total of 33 years of data, from 1985 to 2017. The challenge in applying such detection with Landsat imagery
would be to acquire a dataset of annual time series of cloud- and aerosol-free images for the whole area, and correct for inter-annual variations in the signal.

### 4.4 Bamboo cohort age and reflectance variability

When reconstructing the spectral response of the bamboo-dominated forest as a function of cohort age (Fig. 6), we found that two spectral bands, the NIR-1 and NIR-2, followed our initial assumption of overall reflectance increase with bamboo cohort
age and of sharp decrease at the time of die-off.

Between 1 and 12 years of cohort age, the NIR-1 reflectance did not show a continuous increase (Fig. 6), while it presented strong correlation (r = 0.41) with bamboo-free forest (Fig. 7). The NIR-2 reflectance, however, showed a slight monotonical increase (Fig. 6) with weak correlation (r = 0.19) to bamboo-free forest (Fig. 8). Even though, the accuracy on detecting juvenile bamboos was poor (Fig. 8). Thus, it is very difficult to identify the bamboo dominated patches in this hidden juvenile
age without identifying the prior death event, as reported in a previous study (Carvalho et al., 2013).

The NIR signal suddenly increased at 12-14 years of age, which we believe had two possible explanations. First, there was a change in the density of leafy bamboo branches in the upper forest canopy, where they are visible to the satellite. This is supported by the field observations of Smith and Nelson (2011), in which juvenile bamboo cohorts reach the upper forest canopy by 10 years of age and accelerate in growth due to increased access to solar radiance. They observed that bamboo
density doubled (from 1000 to 2000 culms ha$^{-1}$) and basal area almost tripled (from 2.1 to 5 m$^2$ ha$^{-1}$) between 10 and 12 years of age. The second explanation could be an artifact of our unbalanced sampling for this set of cohort ages. The reflectance values collected for 12–15 years of cohort age were only available from the extremes of our time series (2000 and 2017) due to the bamboo 28 years life cycle and the 18 years of MAIAC data availability.

From 14–27 years, a smooth steady increase occurs only in the NIR-1 signal until the synchronous cohort death, while the
NIR-2 signal seems to have saturated at about age 15 years maintaining a constant signal of 0.3 reflectance until it drops steeply at cohort death. Thus, NIR-1 should present better results for predicting the bamboo age of adult-live stands. Finally, the sharp decrease of NIR-1 and NIR-2 at 28–29 years explain why our bilinear model performed well detecting the time of death. At the time of death, there is a high abundance of dead/dry bamboo branches in the canopy, which reflect less amounts of NIR energy than leafy and photosynthetically active bamboo.



The increases in red reflectance at the die-off, as well as at 1 year of age (Fig. 6A), can also be related to the high abundance of dry bamboo with decreased leaf chlorophyll content and increased non-photosynthetic content. Dry, or dead, vegetation is non-photosynthetically active, and, thus, the incoming red energy near 672 nm is not absorbed by the plant's chlorophyll, that is, causing an increase in the red reflectance (Daughtry, 2000) The dry culms can take up a few years to decompose (Carvalho et al., 2013), which may explain the reason for still observing an increased red signal at 1 year of age.

The large variability in the curves was very likely due to the occurrence of different bamboo abundance and/or forest structure among the area, as well as the inter-annual variability in the signal. Even though, we were able to extract the annual changes in reflectance and predict bamboo ages with 2.92 and 4.25 years RMSE (Fig. 8) using NIR-1 and NIR-2, respectively. The data of each age class was merged from different year composites of the whole time series, thus incorporating the noise in inter-annual variability. Three factors contributing to such noise could be the: (i) temporary formation of green leafy secondary forest, spectrally similar to adult bamboo, in large forest gaps left by the dead bamboo; (ii) semideciduous nature of the trees that are mixed in with bamboo, in the seasonally drier parts of the bamboo range; and (iii) death of bamboo revealed suppressed trees below the bamboo canopy.

### 4.5 Bamboo die-off prediction

By applying the empirical bamboo-age reflectance curves, we estimated the bamboo die-off year for all bamboo patches of the region providing a detailed map of bamboo die-off year and ages (Fig. 8) and bamboo patch size description (Table 1). The estimated die-off events between 2000 and 2017 were similar to the ones detected using the bilinear model. Regarding predictions between 2018 and 2028, the estimate of the exact die-off year was not so accurate (at best 20% accuracy) because those bamboo cohorts were mainly at the juvenile stage during the MODIS (MAIAC) time series period and did not die. However, we believe that the predictions using the NIR-1 were at acceptable levels (RMSE = 2.92 years) when considering that: (i) the Landsat validation points based on visual interpretation can have a deviation of 1 year; and (ii) we assumed that every bamboo cohort had the same life cycle length of 28 years, while we know that it can vary between 27-32 years (Carvalho et al., 2013).

The potential applications of the bamboo die-off year or age map are various. Since areas with dead bamboo are difficult to maintain trails and hinder the work of rubber trappers (Carvalho et al., 2013), it can be used in forest management planning in order to avoid areas where die-off year occurred in the last 3 years and dry culms are still not decomposed, or to avoid areas with likely future die-off. It can also be used for public policy planning regarding food and human health security, for example, in bamboo forests in Southeast Asia, where the bamboo reproductive events cause huge rodent invasion and proliferation that then damage nearby crop plantations (Fava and Colombo, 2017).

### 4.6 Fire occurrence and bamboo

Fire did not occur in all areas where bamboo died (Fig. 5). Hence, we cannot fully support the 'bamboo-fire hypothesis' from Keeley and Bond (1999), because that would require that all bamboo patches burned after die-off Smith and Nelson (2011). We also did not observe an overall increased fire probability over dead than live bamboo in non-drought years. However, our



findings suggest that forests with recently dead bamboo exposed to severe drought are more susceptible to fire occurrence, as there were 45% higher area-normalized mean active fire detections in dead than live bamboo during severe drought years, such as 2005, 2010, 2015 and 2016. When considering the total fire occurrence, we did not observe an overall significant increase in fire occurrence during the 2005 and 2010 major droughts when compared to the other regular years (Brando et al.,
2014). We believe that this is because we filtered the active fire occurring inside the bamboo-dominated areas and pixels with, theoretically, zero non-forested areas using the tree cover products (Hansen et al., 2013), thus excluding the areas of increased fire occurrence in 2005 and 2010 that were reported in the literature (Brando et al., 2014).

Large areas of bamboo die-off occurred close to agricultural lands near Sena Madureira city in the state of Acre, Brazil, during 2015, 2016 and 2017. The combination of increased dry fuel material from bamboos and nearby ignition sources from
land use might have contributed to this increased fire occurrence. This result supports the notion that bamboo die-off enhances fire probability by increasing the dry fuel material in the forest. As we observed in the red wavelength, the reflectance increase was probably associated with greater amounts of dry biomass or non-photosynthetic vegetation in the die-off year and up to 1 year of age (Fig. 6A).

The fire occurrences over bamboo-dominated forests were therefore associated with the proximity to ignition sources, as
less than 1% of forest fire events occurred more than 2 km apart of non-forested areas. This was expected because fire depends on both fuel and ignition to occur. Thus, areas closer to deforested areas, roads and rivers would have higher probability to burn, as probably occurred in 2015, 2016 and 2017. The study of (Kumar et al., 2014) found that 50% of MODIS active fire detections were found within 1 km of roads and rivers, and 95% of the active fires were found within 10 km of roads and rivers in Brazilian Amazon. Fire is known to be associated to deforestation and land use practices in Amazonia such as
slash-and-burn and land preparation, where people remove trees of economic interest and then set the areas on fire in order to clear the land and implement crop plantations or pasture (e.g., Roy et al., 2008). In addition, the fire occurrence beyond 2 km inside the forest was probably underestimated because the forest canopy can obscure fires that occur only on the understorey, and, thus are not detected by the MODIS/Aqua satellite (Roy et al., 2008). Even though, this reinforces a bamboo-human-fire association through the increased land use and cover change. This association is slightly different than it was in pre-Columbian
times (Mcmichael et al., 2014), where geoglyph builders could have used the bamboo die-off patches and fire as an easier way to clear the forest cover to build their monuments, but it should also favor increases in fire occurrence on the vicinities of bamboo-dominated areas, thus leading to potential bamboo expansion.

The higher fire probability in dead bamboo patches during drought events, altogether with the increasing human influence, can favor increases in bamboo abundance and expansion overtime by assisting them in their competition with trees. A previous
study showed that fire favored the *Guadua* bamboo expansion in the region, because the bamboo individuals have faster responses to catastrophic disturbance such as fires than tree species (Smith and Nelson, 2011). Thus, when a fire occurs inside or close to a bamboo forest patch, it may favor the growth of bamboo seedlings – derived from the massive amount of seeds that have been launched during the reproductive phase and prior to death – and the vegetative expansion of the adult bamboo.

Our findings regarding bamboo die-off year being associated to fire occurrence, mainly in drought years, might have impli-
cations to fire control policies, such as in the state of Acre in Brazil, where many bamboo-dominated areas occur nearby human




settlements and that these extreme climate events are occurring within 5 years' interval in Amazonia. By knowing where and when the die-offs are occurring, public policies can be made to avoid fire ignitions in such areas or prepare the fire brigades to attend to potential fires.

## 5 Conclusions

This study demonstrates that the NIR reflectance is more sensitive to the bamboo life cycle than the other spectral intervals and can be used to detect and map bamboo-dominated forests distribution, age structure, and death. The automatic bamboo die-off detection achieved an accuracy above 79% by assessing the point of maximum correlation between the near infrared time series and a bilinear model of linearly increasing NIR with a sharp decrease at the end. After merging the die-off map with the live bamboo map, a total of 155,159 km$^2$ of bamboo-dominated forests was mapped in the region. This area was probably

underestimated due to the limited temporal coverage of the MODIS (MAIAC) time series. The 'bamboo-fire hypothesis' was not fully supported by our results, because most bamboo cohort did not show fire occurrence after the reproductive event and die-off, and, in general, dead bamboo did not show higher fire probability than live bamboo. Nonetheless, when under severe droughts effects, forests with recently dead bamboo are more susceptible to fire than forests with live bamboo, being affected by 45% more fire occurrence. The fire in these areas is mostly associated with ignition sources from land use, suggesting

a bamboo-human-fire association. The interaction of dead bamboos and ignitions cause increased fire occurrence that may contribute to the maintenance of bamboo, burn adjacent forested areas and promote tree mortality, and ultimately the expansion of bamboo into adjacent areas.

Further research of bamboo dynamics can use the current mapping approach with other remote sensing data (e.g. Landsat data with better spatial resolution and longer time series). Using this approach, one can evaluate the temporal dynamics of the

reproductive events (e.g. spreading of flowering waves) and map the bamboo-dominated areas in other parts of the world. Our bamboo-dominated forests and die-off maps can support other studies to understand wildfire dynamics, carbon assimilation in trees and bamboos, tree mortality, fauna/flora demography and species distribution.

*Data availability.* The processed MODIS (MAIAC) data and bamboo maps processed in this paper are freely available and published at: http://doi.org/10.5281/zenodo.1229426.

*Author contributions.* R.D., F.H.W. and L.E.O.C.A. designed the study; R.D. and F.H.W. processed the data and performed the analysis; R.D., F.H.W., L.S.G., B.W.N. and L.E.O.C.A. interpreted the results; R.D. and F.H.W. wrote the manuscript with consultation from L.S.G., B.W.N. and L.E.O.C.A.; All authors provided critical feedback on the paper's discussion and improvement.

*Competing interests.* The authors declare no competing interests.



*Acknowledgements.* R.D. was supported by São Paulo Research Foundation - FAPESP, Brazil, grants 2015/22987-7 and 2017/15257-8.
F.H.W. was supported by São Paulo Research Foundation - FAPESP, FAPESP, Brazil, grants 2015/50484-0 and 2016/17652-9. The funders
had no role in study design, data collection and analysis, decision to publish, or preparation of the manuscript. We thank NASA, and
especially Dr. Yujie Wang and Dr. Alexei Lyapustin, for providing the freely available MODIS (MAIAC) daily dataset. We also thank Dr.

5   Alexei Lyapustin and Dr. Oliver Phillips for insightful comments on early versions of the manuscript.





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
