# Peer review of "Life cycle of bamboo in southwestern Amazon and its relation to fire events"

_Biogeosciences, 2018_

## Referee Comment (RC1) · Anonymous Referee #1 · 13 Jun 2018

This manuscript uses Landsat and MODIS imagery over the MODIS time period (2001 to 2017) to map bamboo patches (living and dead) in the SW Amazon. The authors then estimate patch age based on change over time and test the 'bamboo-fire hypothesis' by comparing presence of dead bamboo to active fire maps from MODIS. Overall I think that this is an interesting and well researched exploration of an important and understudied part of tropical forests - the presence of large patches of bamboo. My main criticism, however, is in the overall clarity of the description of analyses and results - as I note specifically below, there are many places where it is not clear, at least to me, whether the analysis is at the single pixel, pixel over time, or patch scale, what certain terms mean, and how the analyses support or do not support the conclusions. One other general comment is on the use of active fire detection to conclude that 'most

bamboo corhorts did not burn after die-off' (abstract). While this may be the case, this conclusion is based on the assumption that the MODIS Aqua satellite detects 100% of pixels on fire, while in reality it's likely that fire in some pixels was blocked by clouds, was too small to be detected by MODIS, or wasn't burning as the satellite passed overhead. I'm not sure if/how these uncertainties were incorportated into the INPE database, but this source of uncertainty should at least be acknowledged.

Specific comments:

p. 1 lines 1-5: I don't think it's necessary to describe this other study in the abstract. I would just cut the sentences "In southwest Amazon...quantities of necromass." p. 1 line 8: "the fire hypothesis" -> "the bamboo-fire hypothesis" p. 1 line 9: "the MODIS thermal anomalies product" p. 2 line 7: I'm not an expert in Amazon landforms, but I think this should be 'terra firme' throughout (not 'terra fime') - if 'terra fime' is right it probably deserves a short definition since this appears to be an uncommon land type. p. 2 line 20: "In the region" which region is being described here? p. 2 line 22: "in" -> "as" p. 2 line 28: "forming a small" -> "forming small" p. 2 line 30: "maximize once in a lifetime chance..." - I read the Carvalho paper but I still don't totally understand how a temporal offset would maximize the chance of cross pollination. p. 4 line 21: it's helpful to refer to the actual MODIS codes, like MC19A1 (v006, I assume) for consistency p. 4 line 22: Do you actually use all of these bands in the analysis? p. 4 line 28: How did you handle the daily vs 8 day product mismatch? p. 5 line 4: Awesome that this was done in R! Is the code available? section 2.2.2: More detail would be great in this section - did you use 1 image per year? p. 6 line 22: What is a 'percentile' in this context? I've tried pretty hard to figure it out, but I really don't get it, and it's pretty critical to the rest of the manuscript. Is it based on the distribution of values in a pixel? in a patch? This term is also not used in the Carvalho paper. p. 6 lines 26 - 29: What are these distributions telling us? Again, in a given pixel across time? or...? section 2.2.4: as mentioned above, can uncertainty be quantified in the fire data? p. 7 line 13: not sure what Y=x means here. p. 7 line 23: are 'geolocations' the patches of

5 pixels? if there are 390 here, why are there fewer in Fig 4c and d? (I think these should be the same?) p. 7 line 30: "it" = "a bamboo dominated pixel" (I think?) p. 8 line 24: 'geolocations' = 'patches'? pixels? random samples? p. 9 line 27: 'followed a normal distribution (p=0.33)' -> this is a K-S test, right? if yes, 'did not significantly differ from normal' would be more clear, I think. Figure 3 caption: "(hatched)" -> "(hatched in Figure 1)" section 3.2.3: I'm having a hard time grasping exactly how this cohort age analysis using NIR reflectance fits with everything else, especially given that the results differ when different bands are used... (Figure 7) and the accuracy seems low (p 13 line 10)? Is this meaningful? If patches of dead bamboo are being mapped visually, is this fitting necessary to estimate future dieoff? Figure 5: These colors are really hard to see even for a non visually impaired person -> check out colorbrewer2.org for color schemes that are colorblind friendly. p. 17 line 15: 'did show' what? p. 17 line 19: "...in dead and live bamboo" in non drought years? p. 18 line 3: 96.95 to 99.89% of what? p. 18 line 6: "that" -> "where" p. 18 line 9: "The presence of canopy trees could explain why the tree cover is so high." I'm not sure what this is saying that isn't obvious? p. 21 line 29: it seems like there also might be some interesting carbon cycle implications to this work? p. 21 line 32: I don't know if Keeley and Bond would insist on ALL patches burning to confirm the bamboo-fire hypothesis p. 22 line 35: "nearby" -> "near" p. 23 line 11: "not fully supported"? not at all supported, right? I think the uncertainty in the fire observations is an important caveat here, but these results really refute the bamboo-fire hypothesis at least in this setting.

---

## Author Comment (AC1) · 27 Jun 2018

General comment: This manuscript uses Landsat and MODIS imagery over the MODIS time period (2001 to 2017) to map bamboo patches (living and dead) in the SW Amazon. The authors then estimate patch age based on change over time and test the 'bamboo-fire hypothesis' by comparing presence of dead bamboo to active fire maps from MODIS. Overall I think that this is an interesting and well researched exploration of an important and understudied part of tropical forests - the presence of large patches of bamboo. My main criticism, however, is in the overall clarity of the description of analyses and results - as I note specifically below, there are many places where it is not clear, at least to me, whether the analysis is at the single pixel, pixel over time, or patch scale, what certain terms mean, and how the analyses support or do not support the

conclusions. One other general comment is on the use of active fire detection to conclude that 'most bamboo cohorts did not burn after die-off' (abstract). While this may be the case, this conclusion is based on the assumption that the MODIS Aqua satellite detects 100% of pixels on fire, while in reality it's likely that fire in some pixels was blocked by clouds, was too small to be detected by MODIS, or wasn't burning as the satellite passed overhead. I'm not sure if/how these uncertainties were incorporated into the INPE database, but this source of uncertainty should at least be acknowledged.

Response: We thank the reviewer for the fruitful comments and suggestions. We improved the text clarity regarding the units (pixel, patch) and terms, and the description of analysis as specifically pointed out. We have also included some new statistics regarding the omission errors of the presented bamboo die-off detection method as a result of a comment from the reviewer. We agree that the fire dataset (the one we used, but also all MODIS-derived in general) underestimate the total fire occurrence because of its coarse spatial resolution and high cloud cover in Amazon, and, thus, we properly acknowledged that in the discussion. We should miss only 5% of the fire occurrence for fires bigger than 0.09 km$^2$, or approximately 10% of MODIS spatial resolution. The MODIS-INPE fire dataset that we used does not have a source of uncertainty product, but it has been validated in a previous paper (cited in the specific comment below) and showed similar results to a product by NASA-EOS also based on MODIS observations. It also presented fairly good results when compared to a finer scale active fire retrievals from ASTER (30 x 30 m spatial resolution) – detailed in the specific comment below. Nevertheless, we don't believe that the underestimate of total fire frequency has affected the conclusions, because, as pointed out by the reviewer, and acknowledged by us, only a small fraction of bamboo-dominated forests burned during the 16 analyzed years, and dead bamboo did not burn more than live bamboo, so the "bamboo-fire" hypothesis was clearly not supported.

Specific comments:

1- p. 1 lines 1-5: I don't think it's necessary to describe this other study in the abstract.

I would just cut the sentences "In southwest Amazon...quantities of necromass."

Response: We agree with the reviewer. We have shortened the first few sentences to: "Bamboo-dominated forests comprise 1% of the world's forests and 3% of the Amazon forests. The Guadua spp. bamboo that dominate the southwest Amazon are semelparous, so flowering and fruiting occur once in a lifetime before death. These events occur in massive spatially organized patches every 28 years and produce huge quantities of necromass.".

2- p. 1 line 8: "the fire hypothesis" -> "the bamboo-fire hypothesis"

Response: Corrected.

3- p. 1 line 9: "the MODIS thermal anomalies product"

Response: Corrected.

4- p. 2 line 7: I'm not an expert in Amazon landforms, but I think this should be 'terra firme' throughout (not 'terra fime') - if 'terra fime' is right it probably deserves a short definition since this appears to be an uncommon land type.

Response: Yes, you are correct, it was a typing error, we corrected it to "terra firme" in both p. 2 line 7 and p.4 line 7.

5- p. 2 line 20: "In the region" which region is being described here?

Response: It is the southwest Amazon. To improve clarity, we adjusted the sentence to "A total of 74 different bamboo populations, that is, patches having individuals of the same internal age, have been so far identified in the southwest Amazon, with a mean patch area of 330 km$^2$, and up to 2,570 km$^2$ for the largest patch (Carvalho et al., 2013).

6- p. 2 line 22: "in" -> "as"

Response: Corrected.

7- p. 2 line 28: "forming a small" -> "forming small"

Response: Corrected.

8- p. 2 line 30: "maximize once in a lifetime chance..." - I read the Carvalho paper but I still don't totally understand how a temporal offset would maximize the chance of cross pollination.

Response: We re-read the papers that discuss the mast-flowering patches (Franklin, 2004) and realized that this sentence was not making sense with the paragraph idea, which was to give background on flowering waves, dead biomass production, and a brief explanation on why they happen. Thus, we decided to remove the sentence. Franklin, 2004. https://doi.org/10.1111/j.1365-2699.2003.01057.x

9- p. 4 line 21: it's helpful to refer to the actual MODIS codes, like MC19A1 (v006, I assume) for consistency

Response: Agreed. The text was adjusted to: "Daily surface reflectance data were obtained from the MODIS product MCD19A1-C6, acquired from Terra and Aqua satellites, from 2000 to 2017 (Lyasputin and Wang, 2018), corrected for atmospheric effects by the MAIAC algorithm (Lyapustin et al., 2012)." Lyapustin, A., Wang, Y. (2018). MCD19A1 MODIS/Terra+Aqua Land Surface BRF Daily L2G Global 500m, 1km and 5km SIN Grid V006 [Data set]. NASA EOSDIS Land Processes DAAC. doi: 10.5067/MODIS/MCD19A1.006

10- p. 4 line 22: Do you actually use all of these bands in the analysis?

Response: Yes, they were used on empirical bamboo-age reflectance curves analysis (Figure 6) to explore the spectral variation according to bamboo age and demonstrate that NIR band is the most useful to detect die-off.

11- p. 4 line 28: How did you handle the daily vs 8 day product mismatch?

Response: We don't think there is a possible correction to be done here, as we applied

the BRDF correction as it is described in Lyasputin et al. (2012) paper. During the 8-day window, the MAIAC algorithm integrates daily observations with different view angles and retrieve the parameters for BRDF correction of daily observations. The paper report that robust and consistent retrievals are obtained with at least 4 observations. It also tests and corrects the parameters for potential land surface change within the window (Lyasputin et al., 2012). Besides that, variations in sun illumination geometry during the 8-day window are insignificant. We adjusted the sentence in the text to better describe this to the reader: "Parameters of the RTLS model and BRDF kernel weights are part of the MAIAC product suite with a temporal resolution of 8 days – a period which daily observations of different view angles were integrated and used for BRDF parameters retrieval". Lyasputin et al., 2012. https://doi.org/10.1016/j.rse.2012.09.002

12- p. 5 line 4: Awesome that this was done in R! Is the code available?

Response: The MAIAC atmospheric corrections and creation of BRDF parameters were performed and made available by NASA, led by Dr. Alexei Lyasputin. However, the rest of processing (BRDF normalization, composite, mosaic) for the whole South America during 2000-2017 was coded by me, and yes, in R. It was quite a challenge and took some months. The code is available here https://github.com/ricds/maiac_processing. The code is not clean as my specialization is not on programming, but whoever would have the interest to use it to process MAIAC data into composites by himself can contact me and I can help.

13- section 2.2.2: More detail would be great in this section - did you use 1 image per year?

Response: Yes, it was one image per year. In this section, we described the Landsat data that was used in the section 2.3.4 to visual interpret die-off events and validate the prediction model. We adjusted the sentence to make it clearer that we used one image per year: "A time series of Thematic Mapper (TM)/Landsat-5 data was obtained from 1985 to 2000 (one image per year) in order to visually detect die-off events that occurred in the last life cycle of bamboo and validate age predictions – further described in the die-off prediction section.". By the end of the paragraph, we added the information on which scenes (path/row) were analyzed: "The path-row (World Reference System 2) of the time series were: 006-065, 003-066, 002-067, 003-067, 005-067, and 003-068."

14- p. 6 line 22: What is a 'percentile' in this context? I've tried pretty hard to figure it out, but I really don't get it, and it's pretty critical to the rest of the manuscript. Is it based on the distribution of values in a pixel? in a patch? This term is also not used in the Carvalho paper.

Response: We analyzed the 1st, 50th and 99th percentile of tree cover product (Hansen et al., 2013) considering all pixels inside the bamboo map delineated by Carvalho et al. (2013). We reworked the paragraph to improve clarity: "In order to analyze the tree cover variability in forests with and without bamboo, we used the bamboo map from Carvalho et al. (2013) as a mask to analyze the tree cover product (Hansen et al., 2013) considering all pixels inside the bamboo map. This map was obtained in the previous study by visual interpretation of live-adult bamboo using two Landsat mosaics 10 years apart from each other (1990 and 2000), supported by the known locations and dates of five bamboo dominated areas. Considering only the pixels inside the bamboo-dominated map, we calculated the 1st, 50th and 99th percentiles of the tree cover product and generated a map of areas below the 1st, between the 1st and 99th, and above the 99th percentiles of tree cover."

15- p. 6 lines 26 - 29: What are these distributions telling us? Again, in a given pixel across time? or...?

Response: They are telling us about the average, standard deviation and skewness of NIR signal overtime for all pixels in each tree cover percentile class in order to compare the NIR signal between forests with and without bamboo. For normal distribution, the average and standard deviations were calculated. When different than normal,

we applied a more appropriate method to estimate average, standard deviation and skewness parameter (Fernandez and Steel, 1998). As we discussed in the results, for example, if the distribution has a higher NIR average value and is right-skewed, the pixels are likely belonging to bamboo-dominated forests, because of higher NIR values from adult bamboo. We adjusted the text to improve clarity: "In order to compare the NIR signal between forests with and without bamboo, we analyzed the MODIS NIR-1 reflectance for all pixels overtime in the tree cover classes: below 1st, between 1st and 99th, and above the 99th percentile. We tested the distribution of NIR values for normality using a two-sided Kolmogorov-Smirnov test at a 1% significance level. For normal distribution, the average and standard deviation of distributions were computed. For skewed distribution, a more appropriate method was applied to estimate the average, standard deviation and skewness parameter (xi) (Fernandez and Steel, 1998)."

16- section 2.2.4: as mentioned above, can uncertainty be quantified in the fire data?

Response: Unfortunately, the MODIS-INPE active fire dataset we used does not have an uncertainty parameter. However, Morisette et al. (2005) conducted a validation of MODIS active fire retrievals from both (1) NASA EOS and (2) INPE, comparing their results to active fire retrievals from ASTER satellite (finer resolution, 30 x 30 m) and concluded that they were both fairly good. The MODIS-INPE dataset presented high accuracy (95%) for active fires bigger than 0.09 $km^2$, which correspond to 9% of the MODIS spatial resolution. Even though, as the reviewer pointed out in the review introduction, Morisette et al. (2005) highlighted that MODIS active fire detections should be treated as a lower bound of total fire occurrence, as it underestimates small fire occurrences due to the coarse spatial resolution, high cloud cover, and when having high viewing angles (> 15 °). We added this limitation to the discussion in section 4.6, p. 22, line 23: "The MODIS active fire detections should be treated as a lower bound of fire occurrence, as it underestimates fire occurrences, mainly the small ones with less than 0.09 $km^2$, due to the coarse spatial resolution, high cloud cover, and when

having high viewing angles (> 15 °) (Morisette et al., 2005). Morisette et al., 2005. https://journals.ametsoc.org/doi/abs/10.1175/EI141.1

17- p. 7 line 13: not sure what Y=x means here.

Response: It was a failed attempt to describe the linear equation between bamboo age and NIR signal, but we agree that it was confusing and not helpful, so decided to remove it. We reworked the sentence to improve clarity of the bilinear model: "A linearly increasing NIR reflectance vector (1 to 28%) with bamboo age (1 to 28 years), followed by an abrupt NIR decrease to 0% at 29 years of bamboo age."

18- p. 7 line 23: are 'geolocations' the patches of 5 pixels? if there are 390 here, why are there fewer in Fig 4c and d? (I think these should be the same?)

Response: Good question. For each patch (of several pixels), 5 pixels' geolocations were acquired. So, 78 patches equal to 390 pixels/geolocations for validation. Now, there are two explanations: First, if you mean the number of circles in Fig 4c and 4d, it is because we aggregated the samples when they hit the same observed and estimate die-off year. We did this as a way to improve visualization of the agreement, or otherwise, samples would just overlap. To improve clarity, we changed "Samples" to "Pixels", and added this sentence to the caption of Fig 4 (and also Fig 8, which is the prediction): "Size of circles is related to the number of pixels that hit the same observed/estimate die-off year". Second, in order to map the die-off (Fig 4a and 4b), we selected only the pixels with significant relationship with the bilinear model (p < 0.001). When we compared our validation dataset (390 pixels) with the resulting maps, for NIR-1 (Fig 4c) and NIR-2 (Fig 4d) there were actually only 334 and 362 pixels available (p < 0.001), respectively. Thus, a total of 56 and 28 pixels were not classified as die-off, so they were not included in the accuracy assessment. However, now that you pointed this out, we decided to include this information in the results to represent the omission errors of 56/390 = 14.4% and 28/390 = 7.2% for NIR-1 and NIR-2, respectively. When the two maps are merged, the omission error was reduced to 4.1%, while accuracy

was maintained (80%). Thus, we added this sentence after p.11 line 5: "From the 390 pixels in the validation dataset, 334 and 362 pixels were detected as bamboo die-off by the bilinear model (p < 0.001) using the NIR-1 and NIR-2, respectively. The missing 56 (14.4%) and 28 (7.2%) pixels were considered as omission errors for NIR-1 and NIR-2. When we merged the two maps into a single die-off detection map, a total of 374 pixels from the validation dataset were successfully detected, resulting in only 16 (4.1%) missing pixels not detected as bamboo die-off, while accuracy and RMSE were 80% and 0.51 yr, respectively."

19- p. 7 line 30: "it" = "a bamboo dominated pixel" (I think?)

Response: Yes, it is. We rephrased to improve clarity: "We used two assumptions to map the live bamboo. Over the 18 years' period, a live bamboo dominated pixel should present: (i) mean NIR reflectance equal to or greater than the median signal of bamboo-free forests; and (ii) an increasing NIR reflectance over time."

20- p. 8 line 24: 'geolocations' = 'patches'? pixels? random samples?

Response: The multiple terms were indeed confusing. Geolocation and pixels meant the same thing, so we decided to change the term geolocation for pixel in all paper, so it is easier to understand. A total of 2 occurrences were found and adjusted.

21- p. 9 line 27: 'followed a normal distribution (p=0.33)' -> this is a K-S test, right? if yes, 'did not significantly differ from normal' would be more clear, I think.

Response: Yes, corrected.

22- Figure 3 caption: "(hatched)" -> "(hatched in Figure 1)"

Response: Corrected to "(hatched in Figure 2)" as the bamboo area in Figure 1 is not hatched.

23- section 3.2.3: I'm having a hard time grasping exactly how this cohort age analysis using NIR reflectance fits with everything else, especially given that the results differ

when different bands are used...

Response: We believe the cohort analysis was important to improve the understand of remote sensing signal variability with bamboo growth overtime, that is, when the signal changes and why, in order to validate our simple bilinear model that we applied to detect the die-off events. The empirical curves showed the "true" remote sensing signal variation with bamboo age, not only for NIR, but in diverse wavelengths. We extracted the ages using the NIR bands, but we were able to reconstruct the time series of the other bands, which, we believe, is a unique and very interesting result. We discussed the implications of such variations, for example, in the Red band, which is related to chlorophyll content. The first paragraph from section 2.3.3, p.8, l.10, was adjusted to improve the clarity of the analysis: "In order to validate the simple bilinear model that was applied to detect the die-off events and improve the understand of remote sensing signal variability with bamboo growth overtime, that is, when the signal changes and why, we used the die-off map to analyze the remote sensing signal variability. Data from all MODIS bands were extracted using the estimated die-off year with very significant correlation ($p < 0.001$) as a starting point. Bamboo cohort age was then calculated backwards and forwards in time during the 2000-2017 period. Reflectance percentiles (1st, 50th and 99th) per age were calculated obtaining, what we called, empirical bamboo-age reflectance curves."

24- (Figure 7) and the accuracy seems low (p 13 line 10)? Is this meaningful? If patches of dead bamboo are being mapped visually, is this fitting necessary to estimate future dieoff?

Response: In our understanding, the reviewer is commenting on Figure 8, instead of Figure 7, which present the map and accuracy of die-off predictions. We agree with the reviewer that the accuracy on predicting the exact die-off year is fairly low. However, we think that the importance here is that the correlation of predicted and reference die-off is actually moderately strong and statistical significant ($r = 0.41$ and $p < 0.01$ for NIR-1), with RMSE less than 3 years, which, we think, is meaningful. Regarding

the last part of the question, we tested the prediction of future die-off to increase our sampling of die-off areas to test the fire hypothesis. Since MODIS data only span the 2000-2017 period, a big portion of bamboo patches did not undergo die-off during that period and, thus, does not present the decrease in NIR with die-off. Mapping all the die-off patches manually would be time consuming and probably less precise, with bias toward identification of big patches.

25- Figure 5: These colors are really hard to see even for a non visually impaired person -> check out colorbrewer2.org for color schemes that are colorblind friendly.

Response: Agreed and corrected. You can check the adjusted figure in the updated manuscript.

26- p. 17 line 15: 'did show' what?

Response: We complemented with "statistical significance on area-normalized mean active fire detections".

27- p. 17 line 19: "...in dead and live bamboo" in non drought years?

Response: The comparison was between dead and live bamboo in drought years. The sentence was not written correctly, so we adjusted it to: "For severe drought years, the area-normalized active fire detections in 2005 (0.32 and 0.18 fires ha−1), 2010 (0.22 and 0.12 fires ha−1), 2015 (0.35 and 0.20 fires ha−1 ) and 2016 (0.57 and 0.33 fires ha−1 ) over dead and live bamboo, respectively, were not statistically different between the two bamboo life stages (p = 0.127)."

28- p. 18 line 3: 96.95 to 99.89% of what?

Response: Tree cover. We rephrased to improve clarity: "We found that the bamboo-dominated forests had a narrow range of tree cover values (96.95 to 99.89%)".

29- p. 18 line 6: "that" -> "where"

Response: Corrected.

30- p. 18 line 9: "The presence of canopy trees could explain why the tree cover is so high." I'm not sure what this is saying that isn't obvious?

Response: Agreed and removed.

31- p. 21 line 29: it seems like there also might be some interesting carbon cycle implications to this work?

Response: We partly agree, but we are not sure if we should add something to the paper. The bamboo-dominated forests have lower aboveground biomass (AGB) (212 Mg/ha) than dense forests (272 Mg/ha) (e.g. Saatchi, et al., 2007). However, it has more AGB than open forests (200 Mg/ha), probably due to bamboo AGB contributing to that stock. It is interesting that, in this paper, the AGB map shows even lower AGB (100-150 Mg/ha) in bamboo-dominated forests of southwest Amazon. Bamboo may limit aboveground biomass stocks through resources competition and increases in tree mortality (Castro et al., 2013), because of the physical harm it causes on trees (Griscom and Ashton, 2003), while the die-off dynamics may trigger something similar to gap dynamics - because of the suddenly more open canopy and increased sun illumination input. However, we don't expect these dynamics to have implications for carbon cycle in long-term, because the die-off events occur every bamboo cohort life cycle, and, thus, that ecosystem should be already adapted to this. It is expected, though, short-term responses such as pulses of net $CO^2$ emissions after die-off, followed by a period of net C uptake as trees and bamboo grow back. Saatchi, et al. 2007. https://doi.org/10.1111/j.1365-2486.2007.01323.x

32- p. 21 line 32: I don't know if Keeley and Bond would insist on ALL patches burning to confirm the bamboo-fire hypothesis

Response: We haven't considered that before, but we agree. The need for all patches burning is not commented in the Keeley and Bond (1999) paper. What we observed in the results was that the total fire frequency was so low that it wouldn't be feasible that fire should be a driver of bamboo dominance in the study area. We adjusted the

first two sentences in discussion in order to highlight the small magnitude of burning areas compared to the total bamboo area: "Fire occurred only in a small fraction of bamboo-dominated areas during the 16 years of fire analysis (Fig. 5), equivalent to 2371 km$^2$ of burnt area or 0.0955% of the total bamboo area (155,159 km$^2$) burning each year. Besides that, the statistical tests comparing dead and live bamboo fire frequency showed that dead bamboo did not burn more than live bamboo (Fig. 11). Thus, we cannot support the 'bamboo-fire hypothesis' from Keeley and Bond (1999)."

33- p. 22 line 35: "nearby" -> "near"

Response: Corrected.

34- p. 23 line 11: "not fully supported"? not at all supported, right? I think the uncertainty in the fire observations is an important caveat here, but these results really refute the bamboo-fire hypothesis at least in this setting.

Response: Yes, we agree with the reviewer. We adjusted the text to: "The 'bamboo-fire hypothesis' was not supported by our results, because only a small fraction of bamboo areas burned during the analysis timescale, and, in general, bamboo did not show higher fire probability after the reproductive event and die-off." The uncertainties were discussed specifically in the fire section. We believe that even though we have an underestimate of the "true" fire frequency, the observed fire frequency was so small that it shouldn't affect the conclusions.

Please also note the supplement to this comment:
https://www.biogeosciences-discuss.net/bg-2018-207/bg-2018-207-AC1-supplement.pdf

**Supplement:**

[revised manuscript text omitted]
 bamboo, we used the bamboo map from Carvalho et al. (2013) as a mask to analyze the tree cover product (Hansen et al., 2013) considering all pixels inside the bamboo map. This map was obtained in the previous study by visual interpretation of live-adult bamboo using two Landsat mosaics 10 years apart from each other (1990 and 2000), supported by the known locations and dates of five bamboo dominated areas.  Considering only the pixels inside the bamboo-dominated map, we

25  calculated the 1st, 50th and 99th percentiles of the tree cover product and generated a map of areas below the 1st, between the 1st and 99th, and above the 99th percentiles of tree cover.

 In order to compare the NIR signal between forests with and without  bamboo, we analyzed the MODIS NIR-1 reflectance for  all pixels overtime in the tree cover classes: below 1st,

[revised manuscript text omitted]

---

## Referee Comment (RC2) · Anonymous Referee #2 · 28 Jun 2018

General comments: This paper proposes methods to address a very challenging problem in remote sensing - distinguishing between two spectrally similar types of land cover (bamboo forest and bamboo-free forest) using moderate spatial and spectral resolution imagery (MODIS and Landsat). The aim is to map a little-studied, yet spatially expansive, ecosystem in the southwest Amazon - and to establish evidence (or lack thereof) for one hypothesis of bamboo forest establishment and expansion. The study focuses on a fascinating ecosystem, and remote sensing provides the most realistic means of collecting data over such a vast spatial extent in a very remote region. However, I have a number of concerns about the style, methods, and conclusions of the current manuscript, as follows:

1. The methods and results sections include overwhelming detail without clearly outlining goals - both in terms of holistically linking the steps in the methods, and in terms of how the methods relate to broader scientific questions. I am left wondering how the conclusions inform our understanding of the origin and/or biogeochemical processes that maintain/promote expansion of the bamboo forest. 2. The great amount of detail included in the methods swamps any discussion of why specific methods were chosen. As a result, there are many places in the manuscript where the reader may be left feeling that they must take things on faith, or, that the steps presented are the result of a circular logic. 3. Several of the steps in the methods depend on thresholds which are determined from sample means, completely ignoring the impact of spectral (and/or spatial) variability. This topic is briefly addressed in the discussion section, but should be treated more rigorously throughout the methods and analysis sections. In fact, I am left feeling that the authors have failed to demonstrate the practical significance of some of their conclusions because the within group variability seems to overwhelm the between group differences. 4. In a similar vein, because many of the steps in the methods depend on sampling statistics, it is problematic that the sampling unit (point, pixel, polygon?) and sampling protocol (sampling strategy, sample size) remain unspecified in most sections of the paper. 5. Finally, I find it difficult to follow how each step and/or product is validated - in terms of methods, reference datasets, and sampling units and protocols.

Specific examples: 1. The level of detail presented in the methods often seems like a list of what was done, versus a careful retelling of the salient details. For example, the detailed comparisons of NIR1 to NIR2 get confusing and are perhaps unnecessary - an alternative would be to simply state that the two bands were each tested as input data and compared based on some criteria. The NIR 1 was determined to be more useful based on specified criteria. Then the discussion and figures that follow could focus on the results for NIR1 only. But, also see question below. Perhaps more importantly, additional discussion is needed to link specific results to broader scientific questions.

2. Steps described in the methods (and the results) often lack a presentation of the

logic behind why specific methods were tested in the first place (and/or ultimately, selected). A few places where more justification is needed about proposed methods: - Why only test NIR bands (rather than combinations of other bands, standard vegetation indices, etc.)? Why assume a linear model? Why choose thresholds vs. probabilities? How confident are you in Carvalho's estimates? What percent of pixels are highly correlated?

3. The first two conclusions of this study - first, that bamboo-dominated forests have lower tree cover values than bamboo-free forests, and second, that the MODIS NIR values have different distributions over the two forest types - are not supported by convincing evidence. In each case, the differences reported are so small (< 0.1% tree cover and <0.1% reflectance) that I would predict other sources of variability and/or error (e.g., radiometric calibration, sensor signal-to-noise ratio, atmospheric correction uncertainties) might overwhelm these differences.

4. In many cases, the methods lack a description of the sampling procedure employed to inform classification strategies. In all cases where data are sampled for "training" statistics, the authors should provide a concise description of the sampling unit, sampling protocol, and datasets sampled. For example, Section 2.3.1 mentions five pixels, 78 patches, and 380 geolocation points - but I am not sure how these numbers are related, nor what datasets are being compared. Section 2.3.4 mentions different numbers of pixels, patches, and geolocation points, and perhaps uses a different reference dataset? Additionally, no information is provided to place the samples within the larger context of the entire study area - i.e., what percent of pixels (or percent area) of the entire study area is sampled, and how might this impact confidence in the resulting predictions. That is, the predictive model is developed based on a very small sample of the entire study area, but little information is provided to assess what impact this has on the "predictive accuracy" for unknown pixels?

5. Building on the previous point, each validation dataset should be clearly (and consicely) described (perhaps summarized in a table) - and discussions of predictive uncertainty should be included in the results section (not just the discussion section).

Technical corrections/suggestions: Consider refining methods to state the goal of each step first, as well as to identify how each of the steps in the methods sections are related. Currently, the steps are presented in isolation from each other, and in some cases, the almost overwhelming detail makes it difficult to follow how the individual steps are related. Start in the abstract by clearly stating research questions and goals.

Clearly cite previous work and clearly identify which steps were followed in the current study.

Consider including a table to present the imagery used in the analysis, including Landsat WRS-2 and MODIS tile coordinates and image dates.

Check the use of the term "cross-validation."

Limitations of the MODIS active fire detection product are mentioned at the very end of the discussion section - what implications does this have for ecological process? Could it be that the bamboo fire hypothesis is still an open question because we are not measuring understorey fires?

---

## Author Comment (AC2) · 24 Jul 2018

General comments: This paper proposes methods to address a very challenging problem in remote sensing - distinguishing between two spectrally similar types of land cover (bamboo forest and bamboo-free forest) using moderate spatial and spectral resolution imagery (MODIS and Landsat). The aim is to map a little-studied, yet spatially expansive, ecosystem in the southwest Amazon - and to establish evidence (or lack thereof) for one hypothesis of bamboo forest establishment and expansion. The study focuses on a fascinating ecosystem, and remote sensing provides the most realistic means of collecting data over such a vast spatial extent in a very remote region. However, I have a number of concerns about the style, methods, and conclusions of the current manuscript, as follows:

[Figure]

1. The methods and results sections include overwhelming detail without clearly out lining goals - both in terms of holistically linking the steps in the methods, and in terms of how the methods relate to broader scientific questions. I am left wondering how the conclusions inform our understanding of the origin and/or biogeochemical processes that maintain/promote expansion of the bamboo forest.

Response: We thank the reviewer 2 for the comments that led to the improvement of the paper, mainly on increasing the clarity of methods section and discussing important aspects on the validation datasets. Specifically to this comment, we agree with the reviewer 2 that the methods section had too much details, and the goal of each analysis was in general not well described. Therefore, we corrected the methods text to address this problem, providing information on how each analysis is linked to the goals, and why the proposed method was chosen for each analyses. The reviewer 2 can check the changes in the updated manuscript, although some of them were shown in the subsequent comments.

About the conclusions, we tested and rejected the bamboo-fire hypothesis which argues that fire is the driver of bamboo domination in the forest. As pointed out by the reviewer 1 comments in a previous revision, and later acknowledged by us, the evidence we presented in the paper did not support the bamboo-fire hypothesis at all even with the underestimate of fire dataset. This was first stated in p.23 l.30: "We cannot support the 'bamboo-fire hypothesis' from Keeley and Bond (1999) because fire occurred only in a small fraction of bamboo-dominated areas during the 16 years of fire analysis (Fig. 6), equivalent to 2371 km$^2$ of burnt area or 0.0955% of the total bamboo area (155,159 km$^2$) burning each year, and the statistical tests comparing dead and live bamboo fire frequency showed that dead bamboo did not burn more than live bamboo (Fig. 11). Hence, we believe that there should be other explanations for bamboo maintenance in the forest, such as bamboo itself being responsible for its maintenance in the forest due to the damage it causes in the trees while increasing tree mortality (Griscom and Ashton, 2003).".

Then, aspects of the detection were further discussed in p.24 l.11: "The fire occurrence beyond 2 km inside the forest was probably underestimated because the forest canopy can obscure fires that occur only on the understorey, and, thus, are not detected by the MODIS/Aqua satellite (Roy et al., 2008). In addition, the MODIS active fire detections should be treated as a lower bound of fire occurrence, as it underestimates fire occurrences in the order of 5% for small fires with less than 0.09 km$^2$, or 10% of MODIS spatial resolution, due to the coarse spatial resolution, high cloud cover, and when having high viewing angles (> 15 °) (Morisette et al., 2005). Nevertheless, we do not believe this might have an impact on rejecting the 'bamboo-fire hypothesis' due to the minimal fraction of fire occurrences occurring over the large bamboo-dominated forests.".

Besides that, the bamboo die-off maps that were produced in the paper can help future studies addressing the bamboo dynamics processes.

2. The great amount of detail included in the methods swamps any discussion of why specific methods were chosen. As a result, there are many places in the manuscript where the reader may be left feeling that they must take things on faith, or, that the steps presented are the result of a circular logic.

Response: We agree. We have included a flowchart in the manuscript (Figure 1) to give a broad view of the analyses and how the sections interact before going into detail. Also, we have adjusted each paragraph in the methods section by first introducing why the analysis was done, stating the method to perform it and why this method was chosen.

Two examples: (1) In p.7 l.4: "The tree cover product was analyzed considering a pre-existent bamboo-dominated forests map from Carvalho et al. (2013) in order to explore the variability of tree cover in forests with and without bamboo which might help mapping the bamboo-dominated forests. We expect that bamboo-dominated forests present lower tree cover values than bamboo-free forest due to its fast dynamics and

higher mortality (Castro et al., 2013; Medeiros et al., 2013). This map was obtained by visual interpretation of live-adult bamboo using two Landsat mosaics 10 years apart from each other (1990 and 2000), supported by the known locations and dates of five bamboo dominated areas. Considering only the pixels inside the bamboo-dominated map, we calculated the 1st, 50th and 99th percentiles of the tree cover product and generated a map showing the areas below the 1st, between the 1st and 99th, and above the 99th percentiles of tree cover. The map was qualitatively analyzed exploring the areas which each percentile covered.";

(2) In p.7 l.28: "To automatically detect the bamboo die-off from 2001 to 2017 we compared each pixel of MODIS (MAIAC) NIR reflectance time series to a bilinear model using Pearson's correlation and an iterative shift approach. The model consisted in a linear increase in reflectance from 1 to 28% between 1 and 28 years of bamboo age followed by an abrupt decrease to 0% when the die-off occur. The model conception was based on Carvalho et al. (2013) findings which showed that forests with adult bamboo have higher NIR reflectance than forests with juvenile and dead bamboo, or without bamboo, and that bamboo present a life cycle approximate to 28 years. Thus, since not much is known about the spectral behavior of bamboo growth with age, we chose a bilinear model to characterize the bamboo signal change overtime because it was the simplest way to represent the change between life stages. We also assumed the signal coming from the trees as constant over time. Therefore, inter-annual reflectance variations were attributed to structural changes in the canopy related to bamboos. The Pearson's correlation coefficient ($r$) between the NIR reflectance time series and the bilinear model for a given pixel was iteratively tested by shifting the position of the NIR time series inside the bilinear model vector. The position showing the highest $r$ corresponded to the estimated age of that pixel from which the die-off year was retrieved. Only pixels with very significant correlations ($p < 0.001$) were selected. The model was tested with both MODIS (MAIAC) NIR bands: NIR-1 band 2 (841-876 nm) and NIR-2 band 5 (1230-1250 nm). Both bands are sensitive to canopy structure scattering, but NIR-2 is also partially sensitive to leaf/canopy water scattering

(Gao, 1996), so that could lead to a different detection between bands."

3. Several of the steps in the methods depend on thresholds which are determined from sample means, completely ignoring the impact of spectral (and/or spatial) variability. This topic is briefly addressed in the discussion section, but should be treated more rigorously throughout the methods and analysis sections. In fact, I am left feeling that the authors have failed to demonstrate the practical significance of some of their conclusions because the within group variability seems to overwhelm the between group differences.

Response: We agree that the impact of spectral variability was not properly discussed in the paper. We have showed in the empirical bamboo-age spectral curves (Figure 7) that there is a lot of variability in the data. We believe the variability comes mainly from variations in forest structure and bamboo density. However, even though such variability exists, it did not affect the die-off detection, because our method was not based on absolute reflectance values, but on the correlation between the data and a reference, such as the simple bilinear model or the empirical curve. Besides that, even with such huge spectral variability, the method was able to detect the bamboo die-off with great performance (80% accuracy) and, although the predictions for 2017-2028 did not have high accuracy on estimating the exact die-off year, it had acceptable levels of error (around 3 years). We have added this sentence to the discussion in p.23 l.4 to complement the discussion on the large spectral variability: "Nevertheless, because our detection and prediction methods were not based on absolute reflectance values, but on the correlation between the time series and a reference, such as the bilinear model or the empirical curve, we do not believe that the large spectral variability should have a major impact on the detection/prediction."

4. In a similar vein, because many of the steps in the methods depend on sampling statistics, it is problematic that the sampling unit (point, pixel, polygon?) and

sampling protocol (sampling strategy, sample size) remain unspecified in most sections of the paper.

Response: We agree. The reviewer 1 has also pointed the terms (pixel, patch, geolocation, polygon) were confusing. Thus, we have improved the clarity on sampling protocols and terms on methods section. For the first validation dataset (MODIS) for die-off detection during 2001-2017, the new sentence in p.8 l.11 is: "For validation purposes, we compared the detected die-off events with recently dead bamboo areas visually detected in MODIS false color composites (bands 1, 2 and 6 in RGB). In this color composite (Fig. 2), adult bamboo patches show bright green color due to the comparatively higher NIR reflectance, while dead bamboo patches present dark blue/gray color. The visual inspection of bamboo die-off using MODIS and Landsat data was consistent with five bamboo mass flowering events observed in the field (Carvalho et al., 2013). In each of the dead bamboo patches visually detected, the geographic location and die-off year were registered for a sample of 5 random pixels. A total of 78 dead bamboo patches were identified in the 2001–2017 period, thus the validation dataset was composed of 390 pixels with corresponding year of bamboo death - the spatial and temporal distribution of the samples are shown in the supplementary material. For these pixels, the die-off year detected by our model was retrieved and compared to the validation dataset. To assess the detection, we calculated the accuracy (%) on detecting the exact die-off year, Pearson's correlation and p-value, and the root mean square error (RMSE) between the automatically detected and visually interpreted die-off year.".

For the second validation dataset (Landsat) for die-off prediction during 2018-2028, the new sentence in p.9 l.20 is: "Since the validation for 2018-2028 predictions could not be conducted using MODIS data because its time series do not span that time period, we used yearly TM/Landsat-5 color composites (bands 2, 4 and 1 in RGB) during the 1985-2000 period to visually detect the bamboo die-off events that occurred in the last bamboo life cycle and validate the predictions. We assumed that the die-off events

that happened in this period would happen again in the next life cycle of the bamboo, from 2018 to 2028. Therefore, we added 29 years to the visually detected die-off year in order to match the next life cycle. The sampling procedure for the validation dataset was similar to the detection, where 5 pixels were randomly collected for each recently dead bamboo patch visually identified in a given year. A total of 35 dead bamboo patches were identified and 175 pixels were collected with the corresponding years of death. The assessment was conducted by calculating the same metrics as in the die-off detection section. Additionally, in order to assess if the prediction error was randomly distributed, the residuals from predicted minus observed die-off year, where observed is the die-off from the Landsat validation dataset, was tested for normality using a two-sided Kolmogorov-Smirnov test at a 1% significance level."

5. Finally, I find it difficult to follow how each step and/or product is validated - in terms of methods, reference datasets, and sampling units and protocols.

Response: We agree that it was not easy to understand in the way it was pre-sented. There were two validation datasets for two different analyses. One dataset used MODIS data from 2000-2017 and the other used Landsat data from 1985-2000. In both of them, color composites were used to visually inspect die-off events and collect pixels to create validation datasets. The validation dataset from 2000-2017 (MODIS) was used to assess the bamboo-die off detection by the bilinear model running on MODIS time series during 2001-2017. The validation dataset from 1985-2000 (Landsat) was used to assess the predictions of bamboo die-off during 2018-2028 by the model running on MODIS time series and the NIR empirical bamboo-age reflectance curve. This was possible because the die-off events that occurred in the previous life cycle of bamboo (1985-2000) were expected to occur again 28 years later (2018-2028). As we have shown in the last comment, we have improved the description on sampling pixels from bamboo die-off patches for validation datasets. We also believe that the flowchart that we have included (Figure 1) will help the reader

to better understand how each validation dataset was used.

Specific examples:
1. The level of detail presented in the methods often seems like a list of what was done, versus a careful retelling of the salient details. For example, the detailed comparisons of NIR1 to NIR2 get confusing and are perhaps unnecessary – an alternative would be to simply state that the two bands were each tested as input data and compared based on some criteria. The NIR 1 was determined to be more useful based on specified criteria. Then the discussion and figures that follow could focus on the results for NIR1 only. But, also see question below. Perhaps more importantly, additional discussion is needed to link specific results to broader scientific questions.

Response: We agree. We have showed the results for both NIR bands because they mapped slightly different areas of bamboo die-off probably because of the different sensitivities to vegetation composition. This is specially highlighted in the Figure 7 where the NIR-2 remains at its lowest during 0-2 years. Thus, we have included a sentence informing on these differences in mapped areas between NIR-1 and NIR-2 in p.12 l.5: "When comparing the areas detected solely by one of the two bands, NIR-1 detected more pixels towards the end of the time period, i.e. die-off areas from 2017 in the north-east between 8-9°S and 69-70°W, while NIR-2 detected additional pixels in the beginning of the time period, i.e. die-off areas from 2001 in the central region between 9-10°S and 70-71°W.", and adjusted the discussion on why they provided different maps in discussion in p.20 l.21: "When comparing the NIR-1 and NIR-2 bands, the leaf/canopy water sensitivity from NIR-2 might have contributed for a slightly better performance on bamboo die-off detection and the detection of different areas between the bands, which contributed for a larger coverage of the bamboo-dominated forests. This different sensitivity to vegetation structure is specially highlighted in the Figure 7 where the NIR-2 remains at its lowest during 0-2 years, explaining why NIR-2 band maps different areas than NIR-1.".

When we combined both maps, we were able to obtain a map covering a larger area of bamboo die-off with similar accuracy (80%), but lower omission errors (4.1%) than the NIR-1 (14.4%) and NIR-2 (7.2%) maps. We decided to include the combined die-off map to the supplementary material (attached Figure 1 - Bamboo die-off during 2001-2017 from the combined detections using MODIS (MAIAC) NIR-1 and NIR-2 and the bilinear model). Regarding the scientific questions, our evidences clearly do not support the bamboo-fire hypothesis, and, in relation to the die-off patches, we believe that knowing where and when the bamboo dies is an important information for future studies of the bamboo-dominated forests ecosystems. Thus, we added this sentence to discussion in p.23 l.25: "It could also be used to explore broader scientific questions on the ecology of bamboo-dominated forests such as studies on maintenance/expanse of bamboo patches, flowering waves, cross-pollination between patches, fauna habitat dynamics, impacts on short and long-term carbon dynamics.". However, the pursuit of such analyses was out of the scope of this paper.

In order to highlight the best prediction map (NIR-1) we have adjusted the Figure 9 to only include results from the NIR-1 prediction: the map, estimate vs. predicted, and residual error. Thus, the predictions from NIR-2 should be added to supplementary material as shown in attached Figure 4.

2. Steps described in the methods (and the results) often lack a presentation of the logic behind why specific methods were tested in the first place (and/or ultimately, selected). A few places where more justification is needed about proposed methods: 1- Why only test NIR bands (rather than combinations of other bands, standard vegetation indices, etc.)? 2- Why assume a linear model? 3- Why choose thresholds vs. probabilities? 4- How confident are you in Carvalho's estimates? 5- What percent of pixels are highly correlated?

Response: We agree that justifications were missing in the methods. As we stated in previous comments, we have improved the methods description to address

each of the presented problems. Now answering the specific points, which also provide examples of the changes:

(1) We focused the die-off detection on the NIR bands because of two reasons: (i) a previous work from Carvalho et al. (2013) showed that NIR band presented the best separability between bamboo life stages, where higher NIR was observed for adult bamboo than juvenile and dead bamboo; and (ii) we actually did some testing with other bands before the manuscript was written and they provided poor detections; the empirical curves (Fig. 7) corroborate that the other bands does not have a clear change with die-off. When adjusting the methods section, we added this sentence to clarify why we used the NIR band in p.7 l.31: "The model conception was based on (Carvalho et al., 2013) findings which showed that forests with adult bamboo have higher NIR reflectance than forests with juvenile and recently dead bamboo, or without bamboo, and that bamboo present a life cycle approximate to 28 years.". By retrieving the empirical bamboo-age spectral curves, we were able to demonstrate that the remaining spectral bands do not show an abrupt change in reflectance in the die-off year. We did not explore additional attributes such as combinations of bands and spectral indices because our simple model with only the NIR band already presented high accuracy (80%) and very low RMSE error (0.5 RMSE). Even though, we think that should be interesting to use different attributes to potentially improve detection. We added this sentence to conclusions in p.25 l.28: "The mapping approach can be applied with other remote sensing data, such as Landsat data with better spatial resolution and longer time series, and tested with different spectral bands and attributes to further improve the detection."

(2) We used the bilinear model because it was the simplest way to represent the spectral variation changes with bamboo growth overtime as there was no current knowledge on the spectral responses of these bamboos over time. Once we have detected the die-off with this method and reconstructed the empirical spectral variation, we further demonstrated that the relation is not really a linear increase with age, but, in general, more like an abrupt increase when bamboo reach the canopy after 12

years of age, and then an abrupt decrease with die-off (Fig. 7). To improve clarity, we have added this sentence to methods p.8 l.1: "Thus, since not much is known about the spectral behavior of bamboo growth with age, we chose a bilinear model to characterize the bamboo signal change overtime because it was the simplest way to represent the change between life stages."

(3) Regarding the thresholds, in some parts of the paper where we used thresholds, based on statistics that represent our data with high confidence, such as percentiles that capture 99% of the data or that have very low p-value. For example, in the die-off detection we filtered our map based on p-value statistics from the correlation, so only pixels with very high confidence ($p < 0.001$) would be mapped as a die-off. Also, when we extracted the bamboo-free median NIR signal, in order to get the signal from forests without bamboo, we applied the threshold of 99.88% tree cover that excluded at least 99% of the previously known bamboo-dominated areas.

(4) We are confident that the results from Carvalho et al. (2013) are accurate because their validation was made with field campaigns, aerial flights, and independent field data from previous field studies. Hence, it supports that the visual interpretation of bamboo die-off using color composites is indeed associated with the field processes, and, thus is adequate for our validation purposes.

(5) In the second paragraph of section 3.2.1 we have briefly stated that "The correlation coefficients found in the mapped pixels with significant relationship with our bilinear model ($p < 0.001$) were very strong ($r > 0.7$).". That meant that all pixels presented $r > 0.7$. Since more information could be added to it, we extracted additional statistics and adjusted the text in p.13 l.3 to: "The correlation coefficients found in all the mapped pixels with significant relationship with our bilinear model ($p < 0.001$) were strong ($r > 0.7$). More than 50% presented even stronger correlations ($r > 0.8$), and 15% of pixels presented very strong correlation ($r > 0.9$)."

3. The first two conclusions of this study - first, that bamboo-dominated forests have lower tree cover values than bamboo-free forests, and second, that the MODIS

NIR values have different distributions over the two forest types - are not supported by convincing evidence. In each case, the differences reported are so small (< 0.1% tree cover and <0.1% reflectance) that I would predict other sources of variability and/or error (e.g., radiometric calibration, sensor signal-to-noise ratio, atmospheric correction uncertainties) might overwhelm these differences.

Response: We don't agree with Reviewer 2. The tree cover is lower in bamboo-dominated forests than bamboo-free forests as shown for example in the attached Figure 5 (Tree cover over the whole study area and a tree cover percentiles of bamboo forests in a small subset). The differences in the NIR distribution, more thoroughly discussed below, also supports that these differences distinguish the forest types to a certain degree and were useful to help us detect the live bamboo. However, we did not intend that the tree cover would map the bamboo forests, but rather help us to do it with the NIR signal.

Most bamboo-free forests (north east of study area) presented near 100% tree cover, while the bamboo-dominated forests presented values between 96.95 and 99.88%, and only 1% of bamboo-dominated forests showed tree cover of 100%. Even with this small difference in tree cover percentage, it was possible to observe a coincidence between the tree cover percentile map and a previous mapped bamboo-dominated forests in a few areas of the attached Figure 5, which highlight the difference of tree cover in the borders of the bamboo forests. The percentiles were extracted as a way to select the most probable bamboo-free forests and separate from potential bamboo-dominated forests with high confidence. Hence, later on, we used the bamboo-free median signal as part of the live bamboo detection.

Since it was not easy to visualize the difference in the NIR distributions between the tree cover classes in the Figure 4, we have adjusted it in the updated manuscript by separating the three histograms. The mean difference between NIR reflectance in bamboo-dominated forests (mean = 28.7%) and bamboo-free forest (27.3%) is 1.4% and the distributions largely overlap each other. However, the important message

is that the bamboo-dominated forests NIR signal is skewed to the right, because of the adult bamboo have higher NIR values than juvenile bamboo or forests without bamboo, as shown in Fig. 4. Hence, the first rule used to map the live bamboo was "(i) mean NIR reflectance equal to or greater than the median signal of bamboo-free forests". When we added the second rule "(ii) an increasing NIR reflectance over time", the light green pixels in the bamboo spatial distribution (Figure 6) were mapped, which mostly coincided with the remaining non-dead bamboo areas inside the map from Carvalho et al. (2013). This difference in NIR distributions between the three classes supports that differences in tree cover should occur.

4. In many cases, the methods lack a description of the sampling procedure employed to inform classification strategies. In all cases where data are sampled for "training" statistics, the authors should provide a concise description of the sampling unit, sampling protocol, and datasets sampled. For example, Section 2.3.1 mentions five pixels, 78 patches, and 380 geolocation points - but I am not sure how these numbers are related, nor what datasets are being compared. Section 2.3.4 mentions different numbers of pixels, patches, and geolocation points, and perhaps uses a different reference dataset? Additionally, no information is provided to place the samples within the larger context of the entire study area - i.e., what percent of pixels (or percent area) of the entire study area is sampled, and how might this impact confidence in the resulting predictions. That is, the predictive model is developed based on a very small sample of the entire study area, but little information is provided to assess what impact this has on the "predictive accuracy" for unknown pixels?

Response: Agreed. We have improved the description of the sampling as pointed out by reviewer 2, and also reviewer 1 comments. The two paragraphs were previously cited in the general comment 4. In order to show the spatial and temporal distribution of the validation datasets, we intend to add to the supplementary material two figures

which are attached in this document: (i) Figure 2 - Spatial distribution of validation samples obtained from MODIS (2001-2017) imagery in red and Landsat (1985-2000) imagery in blue. The image at background is a false-color composite from MODIS (MAIAC) images of bands 1 (Red), 2 (NIR) and 6 (shortwave infrared), in RGB, respectively, in August 2015; (i) Figure 3 - Temporal distribution of validation samples for bamboo die-off detection (2001-2017) from MODIS imagery and for bamboo die-off prediction (2018-2018) from Landsat imagery.

We agree that a discussion on the representativeness of the validation datasets was missing. We sampled a total of 390 pixels in 78 bamboo patches to validate detections between 2001-2017, and 175 pixels in 35 patches to validate predictions between 2018-2028. We believe that the sampling was at acceptable levels given that the we found a total of 802 patches between 2001-2017 and 778 patches during 2018-2028, which equals to around 10% and 4.5% of the total patches, respectively. It is noted, however, that our visual analysis sampled mostly big patches that died-off, because those were the ones that we could be sure that were attributable as bamboo die-off. This information was added to the discussion in p.21 l.3: "Our validation dataset was composed of 390 pixels visually detected in 78 bamboo patches during 2001-2017. Therefore, we can be confident that the sampling was representative to our study area given that we found 802 patches in the same time period, that is, the sample consisted in around 10% patches. It is noted, however, that our visual analysis mostly sampled big patches that died-off, because those were the ones that we could be sure that were bamboo die-off.", and p.23 l.17: "The validation dataset for the predictions (2017-2028) corresponded to 175 pixels in 35 bamboo patches and represented 4.5% of the 778 bamboo patches predicted for the 2018-2028 time period.".

Our models did not require any training pixels, because they were based on the correlation of the MODIS NIR time series and a reference – such as the bilinear model and the empirical bamboo-age reflectance curves. Therefore, the validation datasets were used as independent sources of validation, providing robust accuracy metrics for the whole map.

5. Building on the previous point, each validation dataset should be clearly (and concisely) described (perhaps summarized in a table) - and discussions of predictive uncertainty should be included in the results section (not just the discussion section).

Response: We agree. We have adjusted the description of each validation dataset (cited in the general comment 4) and added a brief discussion on its representativeness as pointed out in the previous comment (specific comment 4). Nevertheless, we believe that the report of results is appropriate in the results sections, and its interpretations in the discussion section. We have included two figures in supplementary materials (attached Fig 2 and 3) where one can see the spatial and temporal distribution of samples instead of a table, because we believe that would be more useful for the reader.

Technical corrections/suggestions:
1- Consider refining methods to state the goal of each step first, as well as to identify how each of the steps in the methods sections are related. Currently, the steps are presented in isolation from each other, and in some cases, the almost overwhelming detail makes it difficult to follow how the individual steps are related. Start in the abstract by clearly stating research questions and goals.

Response: We agree. We have added a flowchart (Fig 1) to help the reader get a quick grasp of the whole paper before going into detail. Regarding the abstract, we have added a sentence specifying the main goals of the paper, and then described more specifically what was done. The sentences in p.1 l.5: "In this study, our aim is to map the bamboo-dominated forests and test the 'bamboo-fire hypothesis' using satellite imagery. Specifically, we developed and validated a method to map the bamboo die-off and its spatial distribution using satellite-derived reflectance time series from MODIS (MAIAC) and explored the 'bamboo-fire hypothesis' by evaluating

the relationship between bamboo die-off and fires detected by the MODIS thermal anomalies product in the southwest Amazon.".

2- Clearly cite previous work and clearly identify which steps were followed in the current study.

Response: We have improved the description of steps performed in the paper as commented in the previous responses. We believe we have cited all the previous works published in peer-reviewed journals that mapped the bamboo forests in the southwest Amazon, the most important being Nelson et al. (2004) and Carvalho et al. (2013).

3- Consider including a table to present the imagery used in the analysis, including Landsat WRS-2 and MODIS tile coordinates and image dates.

Response: We have prepared a table containing the Landsat images used for validation of bamboo die-off predictions to be included as supplementary material (attached Table 1).

4- Check the use of the term "cross-validation."

Response: Ok, it was incorrectly used in the paper. We have changed the terms "cross-validation" to only "validation".

5- Limitations of the MODIS active fire detection product are mentioned at the very end of the discussion section - what implications does this have for ecological process? Could it be that the bamboo fire hypothesis is still an open question because we are not measuring understorey fires?

**Table 1.** Dates of TM/Landsat-5 images used for validation of bamboo die-off predictions. The date of each image (YYYY-MM-DD) is presented for each path-row (World Reference System 2) in the columns.

| 006-065 | 003-066 | 002-067 | 003-067 | 005-067 | 003-068 |
|---------|---------|---------|---------|---------|---------|
| 1985-06-28 | 1985-07-09 | 1985-09-04 | 1985-08-26 | 1985-07-23 | 1985-07-09 |
| 1986-08-02 | 1986-07-28 | 1986-08-06 | 1986-09-30 | 1986-07-26 | 1986-10-16 |
| 1987-08-05 | 1987-08-16 | 1987-08-25 | 1987-08-16 | 1987-08-14 | 1987-08-16 |
| 1988-08-07 | 1988-07-17 | 1988-08-11 | 1988-08-18 | 1988-07-15 | 1988-06-15 |
| 1989-08-26 | 1989-07-20 | 1989-08-14 | 1989-09-22 | 1989-09-04 | 1989-08-21 |
| 1990-04-23 | 1990-07-07 | 1990-09-18 | 1990-07-23 | 1990-06-19 | 1990-08-24 |
| 1991-06-13 | 1991-07-26 | 1991-07-27 | 1991-07-26 | 1991-07-08 | 1991-07-10 |
| 1992-10-05 | 1992-08-13 | 1992-07-21 | 1992-07-28 | 1992-08-27 | 1992-07-28 |
| 1993-08-05 | 1993-09-01 | 1993-08-25 | 1993-06-13 | 1993-08-14 | 1993-06-13 |
| 1994-07-23 | 1994-07-18 | 1994-07-27 | 1994-07-18 | 1994-06-30 | 1994-07-18 |
| 1995-08-27 | 1995-08-22 | 1995-07-30 | 1995-08-22 | 1995-06-17 | 1995-07-05 |
| 1996-07-12 | 1996-07-23 | 1996-08-01 | 1996-07-23 | 1996-07-05 | 1996-07-23 |
| 1997-09-01 | 1997-07-10 | 1997-07-19 | 1997-07-10 | 1997-07-24 | 1997-08-27 |
| 1998-07-18 | 1998-07-13 | 1998-09-24 | 1998-08-30 | 1998-09-13 | 1998-07-13 |
| 1999-08-06 | 1999-08-01 | 1999-08-10 | 1999-08-17 | 1999-07-30 | 1999-08-17 |
| 2000-10-11 | 2000-07-18 | 2000-07-27 | 2000-07-18 | 2000-09-02 | 2000-09-04 |

Response: We adjusted the fire discussion as a response to Reviewer 1 comments which reinforced that we should fully reject the 'bamboo-fire hypothesis'. The adjusted sentences in p.23 l.30: "We cannot support the 'bamboo-fire hypothesis' from Keeley and Bond (1999) because fire occurred only in a small fraction of bamboo-dominated areas during the 16 years of fire analysis (Fig. 6), equivalent to 2371 km$^2$ of burnt area or 0.0955% of the total bamboo area (155,159 km$^2$) burning each year, and the statistical tests comparing dead and live bamboo fire frequency showed that dead bamboo did not burn more than live bamboo (Fig. 11). Hence, we believe that there should be other explanations for bamboo maintenance in the forest, such as bamboo itself being responsible for its maintenance in the forest due to the damage it causes in the trees while increasing tree mortality (Griscom and Ashton, 2003)". In the updated manuscript, we acknowledged that the fire datasets underestimate the total fire occurrence but it should not affect the conclusions in p.24 l.11: "The fire occurrence beyond 2 km inside the forest was probably underestimated because the forest canopy can obscure fires that occur only on the understorey, and, thus, are not detected by the MODIS/Aqua satellite (Roy et al., 2008). In addition, the MODIS active fire detections should be treated as a lower bound of fire occurrence, as it underestimates fire occurrences in the order of 5% for small fires with less than 0.09 km$^2$, or 10% of MODIS spatial resolution, due to the coarse spatial resolution, high cloud cover, and when having high viewing angles (> 15 °) (Morisette et al., 2005). Nevertheless, we do not believe this might have an impact on rejecting the 'bamboo-fire hypothesis' due to the minimal fraction of fire occurrences osccurring over the large bamboo-dominated forests.".

Please also note the supplement to this comment:
https://www.biogeosciences-discuss.net/bg-2018-207/bg-2018-207-AC2-supplement.pdf

[Figure]

[Figure]

[Figure]

**Fig. 1.** Bamboo die-off during 2001-2017 from the combined detections using MODIS (MAIAC) NIR-1 and NIR-2 and the bilinear model.

74°W    73°W    72°W    71°W    70°W    69°W    68°W

Fig. 2. Spatial distribution of validation samples obtained from MODIS (2001-2017) imagery in red and Landsat (1985-2000) imagery in blue. The image at background is a false-color composite from MODIS [...]

**Fig. 3.** Temporal distribution of validation samples for bamboo die-off detection (2001-2017) from MODIS imagery and for bamboo die-off prediction (2018-2018) from Landsat imagery.

[Figure]

[Figure]

**Fig. 4.** MODIS bamboo die-off prediction map from 2000 to 2028 using the empirical curves of the near infrared 1 (NIR-2) reflectance as a function of bamboo cohort age (a). Validation between predicted [...]

[Figure]

**Fig. 5.** Tree cover over the whole study area and a tree cover percentiles of bamboo forests in a small subset.

**Supplement:**

[revised manuscript text omitted]